# Implicit Bridge Consistency Distillation for One-Step Unpaired Image Translation

## Abstract

Recently, diffusion models have been extensively studied as powerful generative tools for image translation. However, the existing diffusion model-based image translation approaches often suffer from several limitations: 1) slow inference due to iterative denoising, 2) the necessity for paired training data, or 3) constraints from learning only one-way translation paths. To mitigate these limitations, here we introduce a novel framework, called Implicit Bridge Consistency Distillation (IBCD), that extends consistency distillation with a diffusion implicit bridge model that connects PF-ODE trajectories from any distribution to another one. Moreover, to address the challenges associated with distillation errors from consistency distillation, we introduce two unique improvements: Distribution Matching for Consistency Distillation (DMCD) and distillation-difficulty adaptive weighting method. Experimental results confirm that IBCD for bidirectional translation can achieve state-of-the-art performance on benchmark datasets in just one step generation.

## 1 Introduction

Diffusion Models (DMs) (Ho et al., 2020; Song et al., 2021a;b), which learn the score function of clean data, have demonstrated remarkable generation performance through iterative denoising. They have shown superior performance compared to the classical generation models such as Generative Adversarial Networks (GANs) (Goodfellow et al., 2014a), Variational Autoencoders (VAEs) (Kingma & Welling, 2014), etc. Furthermore, DMs have been widely explored across various domains, *e.g.*, text-to-image generation (Rombach et al., 2022), inverse image problems (Chung et al., 2023), image editing (Mokady et al., 2022), and so forth.

Typically, diffusion models (DMs) can be classified into two groups based on the type of governing equation: Stochastic Differential Equations (SDEs) and Probability Flow Ordinary Differential Equations (PF-ODEs). Although PF-ODEs generally require fewer neural function evaluations (NFEs) during sampling compared to SDEs, they still involve numerous iterative steps, leading to slow inference speeds. To address this issue, various techniques have been explored to accelerate the inference speed of DMs. One prominent approach is distillation-based methods, where a student neural network learns the ODE trajectories generated by a pre-trained teacher diffusion model, enabling one-step generation (Salimans & Ho, 2022; Song et al., 2023; Kim et al., 2024b). However, these methods primarily focus on learning deterministic paths from Gaussian noise to specific data distributions, which restricts their applicability to arbitrary distributions, especially in unpaired scenarios.

On the other hand, Schrödinger Bridge (Schrödinger, 1932) offers a promising approach for translating between two arbitrary distributions using entropy-regularized optimal transport. Various methods have been developed for translating between data distributions, such as those proposed in (Wang et al., 2021; Chen et al., 2021; Liu et al., 2022), though many of these methods are limited to paired settings. In contrast, DDIB (Su et al., 2023) addresses image-to-image translation by concatenating the ODE trajectories of two distinct DMs, making it suitable for unpaired settings, yet it still relies on numerous iterative steps. More recently, UNSB (Kim et al., 2024a) has been introduced to directly tackle unpaired image-to-image translation by regularizing Sinkhorn paths. However, UNSB faces limitations due to its dependence on multiple iterative steps, unidirectional translation, and the use of adversarial training with a discriminator, which can lead to training instability.

| Model | One-Step | Unpaired | Bi-direction | Discriminator |
|---|---|---|---|---|
| SDEdit | ✗ | ✓ | ✗ | ✗ |
| ILVR | ✗ | ✓ | ✗ | ✗ |
| EGSDE | ✗ | ✓ | ✗ | ✗ |
| SDDM | ✗ | ✓ | ✗ | ✗ |
| CycleDiffusion | ✗ | ✓ | ✓ | ✗ |
| DDIB | ✗ | ✓ | ✓ | ✗ |
| DDBM | ✗ | ✗ | ✓ | ✗ |
| UNSB | ✗ | ✓ | ✗ | ✓ |
| **IBCD (Ours)** | ✓ | ✓ | ✓ | ✗ |

Table 1: A systematic comparison of IBCD with other diffusion-based image-to-image translation models highlights several key advantages of our approach.

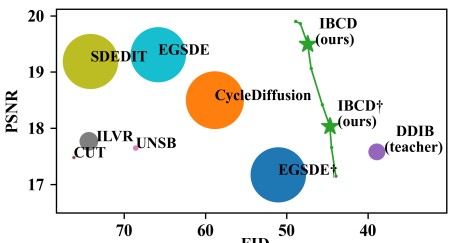

Figure 1: PSNR-FID trade-off comparison with baselines on the Cat→Dog (256) task. The size of the marker represents the NFE.

To address the limitations of the existing method, we aim at the development of a bidirectional one-step generator that enables translation between two arbitrary distributions in unpaired settings without relying on adversarial losses (see a comparison in Table 1). To achieve this, we propose Implicit Bridge Consistency Distillation (IBCD), an extension of the concept of consistency distillation (CD) that incorporates a diffusion implicit bridge model for translating between arbitrary data distributions. Unlike CD, which learns paths from Gaussian noise to data, IBCD connects trajectories from one arbitrary distribution to another one using a Probability Flow Ordinary Differential Equation (PF-ODE), allowing for flexible and efficient distribution translation.

However, simply extending CD can result in reduced distillation efficacy due to error accumulation, as well as challenges related to model capacity and training, which arise from integrating two trajectories and introducing bidirectionality. To address this, we propose a regularization method called Distribution Matching for Consistency Distillation (DMCD). Furthermore, we introduce a novel weighting scheme based on distillation difficulty, which applies a stronger DMCD penalty specifically to samples where the consistency loss alone proves insufficient. By integrating these advanced components, along with an additional cycle loss, our approach significantly enhances the reality-faithfulness trade-off, achieving state-of-the-art performance in a single step, as shown in Figure 1. The main contributions of our work are as follows:

- We propose a novel unpaired image-to-image translation framework, termed Implicit Bridge Consistency Distillation (IBCD), which enables bidirectional translation using only a single neural function evaluation (NFE), achieving state-of-the-art performance on benchmark datasets.
- We introduce additional improvements, including Distribution Matching for Consistency Distillation (DMCD) and an adaptive weighting scheme based on distillation difficulty, to effectively mitigate distillation errors inherent in the process. Additionally, the incorporation of cycle-loss further enhances image translation performance, resulting in more accurate and reliable translations.

## 2 PRELIMINARIES

### 2.1 IMAGE TO IMAGE TRANSLATION WITH DIFFUSION MODELS

**Diffusion Models (DM).** In DMs (Ho et al., 2020; Song et al., 2021b), the predefined forward process with the time variable $t \in [0, T]$ progressively corrupts data into pure Gaussian noise over a series of steps $T$. Specifically, given a data distribution $\mathbf{x}_0 \sim p(\mathbf{x}_0) := p_{\text{real}}(\mathbf{x})$, the distribution $\mathbf{x}_T \sim p(\mathbf{x}_T)$ approaches an isotropic normal distribution as noise is added according to the process $p(\mathbf{x}_t \mid \mathbf{x}_0) = \mathcal{N}(\mathbf{x}_0, t^2\mathbf{I})$. The reverse of this process can be described by an SDE or a PF-ODE (Song et al., 2021b) as follows:

$$\frac{d\mathbf{x}_t}{dt} = -t\nabla_{\mathbf{x}_t} \log p(\mathbf{x}_t) = \frac{\mathbf{x}_t - \mathbb{E}[\mathbf{x}_0|\mathbf{x}_t]}{t}, \tag{1}$$

where the second equality follows from Tweedie's formula, $\mathbb{E}[\mathbf{x}_0|\mathbf{x}_t] = \mathbf{x}_t + t^2\nabla_{\mathbf{x}_t} \log p(\mathbf{x}_t)$ (Efron, 2011; Kim & Ye, 2021). In practice, the neural network is trained to approximate the ground truth

score function $s_\phi(\mathbf{x}_t, t) \approx \nabla_{\mathbf{x}_t} \log p(\mathbf{x}_t)$ or the denoiser $D_\phi(\mathbf{x}_t, t) \approx \mathbb{E}[\mathbf{x}_0|\mathbf{x}_t]$ by denoising score matching (Vincent, 2011). By substituting the trained neural networks into Eq. (1), we can obtain the denoised sample by numerically integrating from $T$ to 0:

$$\mathbf{x}_0 = \mathbf{x}_T + \int_T^0 -t \cdot s_\phi(\mathbf{x}, t)\, dt = \mathbf{x}_T + \int_T^0 \frac{\mathbf{x}_t - D_\phi(\mathbf{x}_t, t)}{t}\, dt, \tag{2}$$

To solve Eq. (2), an ODE solver, denoted as $\texttt{Solver}(\mathbf{x}_T; \phi, T, 0)$ (with an initial state $\mathbf{x}_T$ at time $T$ and ending at time 0, DM parameterized by $\phi$) can be applied. Examples include the Euler solver (Song et al., 2021b; Ho et al., 2020), DPM-solver (Lu et al., 2022), or the second-order Heun solver (Karras et al., 2022). The sampling process typically requires dozens to hundreds of neural function evaluations (NFE) to effectively minimize discretization error during ODE solving.

**Dual Diffusion Implicit Bridge (DDIB).** DDIB (Su et al., 2023) is a simple yet effective method for image-to-image translation that leverages the connection between DMs and Schrödinger bridge problem (SBPs), where DMs act as implicit optimal transport models. DDIB requires training two individual DMs for the two domains A and B, denoted as $s_{\phi^a}$ and $s_{\phi^b}$. The sampling process involves sequential ODE solving as follows:

$$\mathbf{x}^l = \texttt{Solver}(\mathbf{x}^a; \phi^a, 0, T), \quad \mathbf{x}^b = \texttt{Solver}(\mathbf{x}^l; \phi^b, T, 0). \tag{3}$$

Here, $\mathbf{x}^l$ represents the latent code in the pure noise domain, $\mathbf{x}^a$ is the image in the source domain, and $\mathbf{x}^b$ is the estimated image in the target domain. Thanks to the intermediate Gaussian distribution, DDIB automatically satisfies the cycle consistency property without requiring any explicit regularization term (Zhu et al., 2017; Choi et al., 2018).

### 2.2 ONE-STEP ACCELERATION OF DIFFUSION MODELS WITH DISTILLATION

**Consistency Distillation (CD).** The aim of the consistency distillation (CD) (Song et al., 2023) is to learn the direct mapping from noise to clean data. Specifically, the model is designed to predict $f_\theta(\mathbf{x}_t, t) = \mathbf{x}_0$, and is constrained to be *self-consistent*, meaning that outputs should be the same for any time point input within the same PF-ODE trajectory, *i.e.*, $f(\mathbf{x}_t, t) = f(\mathbf{x}_{t'}, t')$ for all $t, t' \in [\epsilon, T]$, with the boundary condition $f_\theta(\mathbf{x}_\epsilon, \epsilon) = \mathbf{x}_\epsilon$. Here, $\epsilon$ is a small positive number, to avoid numerical instability at an $t = 0$. By discretizing the time interval $[\epsilon, T]$ into $N - 1$ sub-interval with boundaries $t_1 = \epsilon < t_2 < \cdots < t_N = T$, the resulting objective function for CD is given by:

$$\mathcal{L}_{\text{CD}}(\theta; \phi) = \mathbb{E}[\lambda(t_n) d(f_\theta(\mathbf{x}_{t_{n+1}}, t_{n+1}), f_{\theta^-}(\hat{\mathbf{x}}_{t_n}, t_n))], \quad n \sim \mathcal{U}[1, N-1]. \tag{4}$$

where $\lambda(t_n)$ is weight hyperparameter, $d(\cdot, \cdot)$ measures the distance between two samples. $\theta^-$ is the exponential moving average (EMA) of the student parameter $\theta$, and $\phi$ represents the pre-trained teacher model, and $\mathcal{U}[\cdot]$ refers to the uniform distribution. The target $\hat{\mathbf{x}}_{t_n}$ is obtained by solving one-step ODE solver, *i.e.*, $\hat{\mathbf{x}}_{t_n} = \texttt{ODESolve}(\mathbf{x}_{t_{n+1}}; \phi, t_{n+1}, t_n)$, from $\mathbf{x}_{t_{n+1}} \sim \mathcal{N}(\mathbf{x}, t_{n+1}^2 \mathbf{I})$.

**Distribution Matching Distillation (DMD).** To distill the diffusion model $s_\phi^{\text{real}}$ into a one-step generator $f_\theta(\mathbf{x}_T) = \mathbf{x}_0$, Yin et al. (2024) proposed DMD to minimize the Kullback-Leibler (KL) divergence between the real data distribution, $p^{\text{real}}$, and the student sample distribution, $p_\theta^{\text{fake}}$. Specifically, DMD introduces an auxiliary *fake* DM, denoted as $s_\psi^{\text{fake}}$, which serves to approximate the score function of the student-generated sample distribution – one that is otherwise directly inaccessible. This estimator is trained concurrently using denoising score matching, dynamically adapting in real-time to the evolving sample outputs as the student model progresses through training. The gradient of the Distribution Matching Distillation (DMD) loss can then be approximated as the difference between the two score functions:

$$\nabla_\theta D_{\text{KL}}(p_\theta^{\text{fake}} || p^{\text{real}}) \approx \nabla_\theta \mathcal{L}_{\text{DMD}} = \underset{\mathbf{x}_t, t, \mathbf{x}_T}{\mathbb{E}} [w_t(s_\psi^{\text{fake}}(\mathbf{x}_t, t) - s_\phi^{\text{real}}(\mathbf{x}_t, t)) \nabla_\theta f_\theta(\mathbf{x}_T)] \tag{5}$$

where $\mathbf{x}_t \sim \mathcal{N}(f_\theta(\mathbf{x}_T), t^2 \mathbf{I})$, $t \sim \mathcal{U}(T_{\min}, T_{\max})$, $\mathbf{x}_T \sim \mathcal{N}(\mathbf{0}, T^2 \mathbf{I})$ and $w_t$ is a scalar weighting factor. DMD serves as an effective distillation loss that optimizes the student model from the perspective of the distribution, without the need to rely on the instability associated with adversarial loss (Goodfellow et al., 2014b).

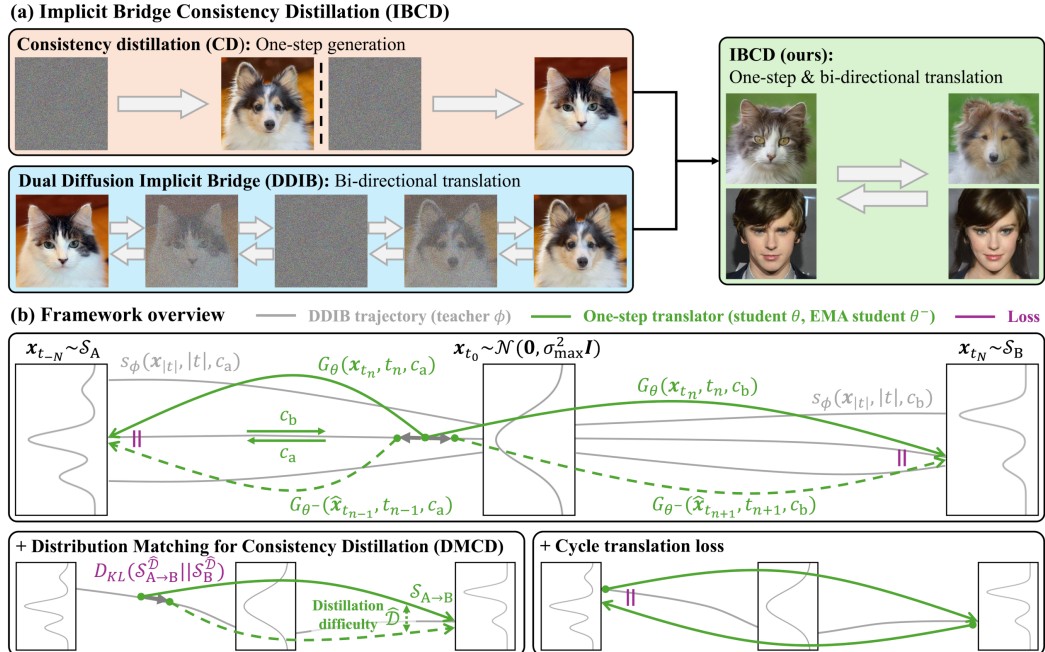

Figure 2: (a) IBCD performs one-step bi-directional translation using a distillation framework that extends consistency distillation with a diffusion implicit bridge. (b) The IBCD framework bridges two distributions by connecting the PF-ODE paths of two pre-trained diffusion models through bidirectionally extended consistency distillation. To mitigate distillation errors, we introduce distribution matching for consistency distillation and a cycle loss.

## 3 METHODS

Building upon existing approaches, our goal is to develop a distillation method for a one-step model that enables bidirectional mapping between arbitrary probability distributions in an unpaired setting, by leveraging pre-trained diffusion models. Specifically, given two domains $\mathcal{X}_A, \mathcal{X}_B$, and unpaired datasets $\mathcal{S}_A = \{\mathbf{x}^a \in \mathcal{X}_A\}, \mathcal{S}_B = \{\mathbf{x}^b \in \mathcal{X}_B\}$, our translator $f_\theta$ is designed to perform two translation functions: $f_\theta(\mathbf{x}^a, c_b) : \mathcal{X}_A \to \mathcal{X}_B$ and $f_\theta(\mathbf{x}^b, c_a) : \mathcal{X}_B \to \mathcal{X}_A$, where $c_a$ and $c_b$ represent class embeddings for the target translation domain. In the following, we introduce our framework for distilling bidirectional DDIB into a unified one-step model in an unpaired setting and then discuss the associated challenges. To address these, we present a novel distillation approach by incorporating distribution matching with a new adaptive weighting factor and a cycle loss to enable bidirectional reconstruction.

### 3.1 IMPLICIT BRIDGE CONSISTENT DISTILLATION

**Definition.** Our model architecture and diffusion process are based on the PF-ODE using EDM (Karras et al., 2022). To handle both domains with one generator, a pre-trained class conditional DMs, $s_\phi(\mathbf{x}_t, t, c)$, is jointly trained for each domain with class conditions $c_a$ and $c_b$. Specifically, the teacher model $s_\phi$ is trained using denoising score matching (DSM) for continuous-time $t = \sigma \sim$ Lognormal $\in (0, \infty)$ without any modification from EDM. The timestep discretization for the sampling process is defined as $[t_0, t_1, \cdots, t_i, \cdots, t_N] = [\sigma_{\max}, \sigma_{\max-1}, \cdots, \sigma_{\min}, 0]$. Since DDIB concatenates two independent ODEs into a single ODE, duplicated timesteps must be redefined for consistency distillation (CD). We introduce a unique discretized timestep index $i$ and redefine the timestep $t$ for the concatenated trajectory ($\mathcal{X}_A \leftrightarrow \mathcal{X}_B$) as follows:

$$i = [\underbrace{-N, -N+1, \cdots, -1}_{\mathcal{X}_A}, \underbrace{0}_{\mathcal{X}_A \cap \mathcal{X}_B}, \underbrace{1, \cdots, N-1, N}_{\mathcal{X}_B}]$$

$$t_i = \sigma_i = [-0, -\sigma_{\min}, \cdots, -\sigma_{\max-1}, +\sigma_{\max}, +\sigma_{\max-1}, \cdots, +\sigma_{\min}, +0] \qquad (6)$$

**Boundary Condition.** Given that the output of the student model is enforced to be *self-consistency* with respect to the timesteps in Eq. (6), we define the student as $f_\theta(\mathbf{x}_t, t, c)$, where $t$ is a non-zero real-valued timestep and $c \in \{c_a, c_b\}$ represents the target domain condition. For simplicity, we define the superscript $'$ as the opposite class embedding, *i.e.*, when $c = c_b$, $c' := c_a$. To enable the bidirectional translation, we redefine the boundary condition of IBCD to depend on the target domain condition $c$:

$$f\big(\mathbf{x}_{\epsilon(c)}, \epsilon(c), c\big) = \mathbf{x}_{\epsilon(c)}, \quad \text{where } \epsilon(c) = \begin{cases} t_{-N+1} &= -\sigma_{\min}, & \text{for } c = c_a \\ t_{N-1} &= +\sigma_{\min}, & \text{for } c = c_b \end{cases}. \tag{7}$$

This boundary condition, alongside the IBCD loss introduced later, ensures that we can translate samples by injecting the desired domain condition: $f(\mathbf{x}_{\epsilon(c)}, \epsilon(c), c') = \mathbf{x}_{\epsilon(c')}$. Specifically, $f(\mathbf{x}_t, t, c_b)$ transforms $\mathbf{x}_t$ at any $t$ between $\mathcal{X}_A$ and $\mathcal{X}_B$ into a clean domain $\mathcal{X}_B$ image $\mathbf{x}_{t_{N-1}}$ belonging to the same ODE trajectory, and vice versa. Since EDM/CD is not defined for negative $t$ values and is not directly aligned with our new boundary conditions, we extended the EDM/CD formulation and applied it to the student model[1]. For further details on this extension, please refer to Appendix B.

**Implicit Bridge Consistency Distillation (IBCD).** To generate data pairs $(\mathbf{x}_{t_1}, \hat{\mathbf{x}}_{t_2})$ that belong to the same PF-ODE trajectory for IBCD, we perform forward diffusion on the dataset and predict the next data point one step ahead using a suitable teacher model and ODE solver. For simplicity, we denote the teacher model $\phi$ conditioned on class $c$ as $\phi^c$. The data pair generation process in the direction of $\mathcal{X}_A \to \mathcal{X}_B$ (*i.e.* $c = c_b$) for each domain is as follows:

$$\hat{\mathbf{x}}_{t_{n_a+1}} = \texttt{Solver}(\mathbf{x}_{t_{n_a}}; \phi^a, |t_{n_a}|, |t_{n_a+1}|), \ \hat{\mathbf{x}}_{t_{n_b+1}} = \texttt{Solver}(\mathbf{x}_{t_{n_b}}; \phi^b, |t_{n_b}|, |t_{n_b+1}|), \tag{8}$$

where $n_a \sim \mathcal{U}[-N+1, -1], n_b \sim \mathcal{U}[0, N-2], \mathbf{x}_{t_{n_a}} \sim \mathcal{N}(\mathbf{x}^a, t_{n_a}^2 \mathbf{I}), \mathbf{x}_{t_{n_b}} \sim \mathcal{N}(\mathbf{x}^b, t_{n_b}^2 \mathbf{I})$. Similarly, in the direction $\mathcal{X}_B \to \mathcal{X}_A$ (*i.e.* $c = c_a$), the data pair for each domain can be generated as:

$$\hat{\mathbf{x}}_{t_{n_a-1}} = \texttt{Solver}(\mathbf{x}_{t_{n_a}}; \phi^a, |t_{n_a}|, |t_{n_a-1}|), \ \hat{\mathbf{x}}_{t_{n_b-1}} = \texttt{Solver}(\mathbf{x}_{t_{n_b}}; \phi^b, |t_{n_b}|, |t_{n_b-1}|), \tag{9}$$

where $n_a \sim \mathcal{U}[-N+2, 0], n_b \sim \mathcal{U}[1, N-1]$. Given these distillation targets, our objective function of IBCD is defined as follows:

$$\mathcal{L}_{\text{IBCD}}(\theta; \phi) = \mathbb{E}_{\mathbf{t}_1, \mathbf{x}_{\mathbf{t}_1}, c}[\lambda(\mathbf{t}_2) d(f_\theta(\mathbf{x}_{\mathbf{t}_1}, \mathbf{t}_1, c), f_{\theta^-}(\hat{\mathbf{x}}_{\mathbf{t}_2}, \mathbf{t}_2, c))], \tag{10}$$

$$\text{where} \quad \mathbf{x}_{\mathbf{t}_1} = [\mathbf{x}_{t_{n_a}}; \mathbf{x}_{t_{n_b}}], \ \hat{\mathbf{x}}_{\mathbf{t}_2} = [\hat{\mathbf{x}}_{t_{n_a\pm1}}; \hat{\mathbf{x}}_{t_{n_b\pm1}}], c \in \mathcal{U}[\{c_a, c_b\}]$$

$$\mathbf{t}_1 = [t_{n_a}; t_{n_b}], \ \mathbf{t}_2 = [t_{n_a\pm1}; t_{n_b\pm1}], \theta^- = \texttt{sg}(\mu\theta^- + (1-\mu)\theta)$$

where $n_{(\cdot)\pm1}$ denotes time index for each distillation direction in Eqs. (8), (9) and $\texttt{sg}$ indicates the stop-gradient operator. For a detailed explanation, see Algorithm 1.

Note that employing a single domain-independent teacher model, rather than two separate models, not only reduces memory consumption but also provides an effective initializer for the student model, serving as an integrated model for both domains. By sharing the class condition in the teacher model and the target domain condition in the student model as a unified embedding, we can effectively utilize the student's initialization weights, since $f(\mathbf{x}_t, t, c)$ is formulated to output a clean image corresponding to domain $c$. This approach distinguishes itself from other methods in the literature (Kim et al., 2024b; Li & He, 2024), which extend CD in both directions or specify a target timestep, without fully integrating the domain conditions into a cohesive framework.

## 3.2 LOSS FUNCTION FOR IMPLICIT BRIDGE CONSISTENCY DISTILLATION

While our IBCD framework facilitates one-step bidirectional transport of the student model in unpaired settings, it faces certain challenges. First, the consistency loss relies on a local consistency strategy (categorized by Kim et al. (2024b)), which aligns consistency only between adjacent timesteps by recursively using the student's output. This can lead to the accumulation of local errors, resulting in a growing discrepancy between the student's prediction $f_\theta(\mathbf{x}_t, t, c)$ and the true boundary value $\mathbf{x}_{\epsilon(c)}$ as the distance from the boundary condition timestep increases. This issue is particularly pronounced in IBCD due to its doubled trajectory length compared to standard CD.

---

[1]Note that this formulation is applied exclusively to the student model.

Second, the student not only has to perform a bidirectional task but also has to learn two different ODE trajectories. Considering that the two ODEs in the teacher model share time steps and are separated only by conditions, this increased complexity can not only affect model capacity but also make training more difficult. The degradation of consistency distillation performance due to the addition of bidirectional features has also been reported by Li & He (2024). Third, unlike EGSDE (Zhao et al., 2022), which can freely adjust the trade-off between reality and faithfulness by weighting expert contributions, vanilla IBCD lacks an explicit mechanism to control this balance, potentially limiting its ability to adapt to diverse scenarios.

**Distribution Matching for Consistency Distillation (DMCD).** To address these issues, we propose Distribution Matching for Consistency Distillation (DMCD), which extends the distribution matching loss to fit within the consistency distillation framework. DMCD builds on the DMD loss by optimizing the KL divergence between the student's output samples and the target domain data distributions across all timesteps in bidirectional tasks. Furthermore, it incorporates the distillation difficulty adaptive weighting factor $\hat{\mathcal{D}}(\cdot, \cdot)$. This adaptive weighting scheme helps to focus the optimization on challenging samples, thereby enhancing the overall performance and stability of the student model during training. The resulting DMCD is given by:

$$\nabla_\theta \mathcal{L}_{\text{DMCD}} = \mathop{\mathbb{E}}_{\mathbf{t}_1, \mathbf{x}_{\mathbf{t}_1}, c, i, \mathbf{x}_{t_i}} [w_{t_i} \hat{\mathcal{D}}(\text{sg}(\mathbf{x}_{\mathbf{t}_1}), c)(s_\psi(\mathbf{x}_{t_i}, t_i, c) - s_\phi(\mathbf{x}_{t_i}, t_i, c))\nabla_\theta f_\theta(\mathbf{x}_{\mathbf{t}_1}, \mathbf{t}_1, c)] \quad (11)$$

$$\text{where} \quad i \sim \mathcal{U}[0, N-1], \ \mathbf{x}_{t_i} \sim \mathcal{N}(f_\theta(\mathbf{x}_{\mathbf{t}_1}, \mathbf{t}_1, c), t_i^2\mathbf{I})$$

where $\mathbf{t}_1, \mathbf{x}_{\mathbf{t}_1}, c$ are defined per from Eq. (10), and $w$ represents a time-dependent weighting factor introduced in DMD. The term $s_\psi(\mathbf{x}_t, t, c)$ denotes a class-conditional fake diffusion model (DM), which is continuously trained via denoising score matching on outputs of student $f_\theta$, adapting as the training progresses. Unlike DMD, DMCD functions as a regularizer rather than the main loss. This distinction is crucial in unpaired settings, where relying solely on the DMCD loss does not ensure a proper connection between two domains. IBCD bridges a trajectory between two distributions using consistency loss, while DMCD addresses the distribution matching issue, working as a loss to increase the reality of the results. This integration allows for improved performance and stability without the drawbacks associated with adversarial training (Zhu et al., 2017; Parmar et al., 2024; Kim et al., 2024a).

**Distillation Difficulty Adaptive Weighting.** DMCD effectively brings the translated distribution closer to the target data distribution, enhancing the reality of the generated samples. However, this can also cause a divergence from the teacher model's estimations, thereby reducing faithfulness to the source distribution. Ideally, DMCD should be applied more intensively to challenging PF-ODE trajectories that the student model struggles to translate accurately, particularly those involving source domain data points near the decision boundary of the source domain. To address this, we propose a *distillation difficulty adaptive weighting* strategy. To quantify the difficulty, we introduce the concept of *distillation difficulty*, $\mathcal{D}([\mathbf{x}_{t_{-N+1}}, \cdots, \mathbf{x}_{t_{N-1}}], c) := d(f_\theta(\mathbf{x}_{\epsilon(c')}, \epsilon(c'), c), \mathbf{x}_{\epsilon(c)})$, which measures the challenge of distilling a given ODE trajectory generated by the teacher between domains. This approach ensures that DMCD is applied more aggressively to the most difficult trajectories, improving the overall translation performance by focusing on areas where the student model needs it most. Such a strategy could strike a balance between source faithfulness and reality by applying the DMCD loss forcefully only to trajectories where the IBCD loss alone is insufficient. However, estimating $\mathbf{x}_{\epsilon(c)}$ and $\mathbf{x}_{\epsilon(c')}$ from a given $\mathbf{x}_t$ using the ODE solver requires at least $N$ NFEs with the teacher model for each DMCD loss calculation, which is computationally impractical. To address this, we propose a one-step approximation of the weighting factor $\mathcal{D}(\cdot, \cdot)$, defined as follows:

$$\hat{\mathcal{D}}(\mathbf{x}_{\mathbf{t}_1}, c) = g(d(f_\theta(\mathbf{x}_{\mathbf{t}_1}, \mathbf{t}_1, c), \ f_{\theta^-}(\hat{\mathbf{x}}_{\mathbf{t}_2}, \mathbf{t}_2, c)))) \quad (12)$$

where $\mathbf{t}_1, \mathbf{t}_2, \mathbf{x}_{\mathbf{t}_1}, \hat{\mathbf{x}}_{\mathbf{t}_2}$ are defined in Eqs. (10), (11) and $g$ is any monotone increasing function. The validity of the alternative weighting factor will be confirmed later through experiments.

**Cycle Translation Loss.** Similar to DDIB, our framework is designed to perform cycle translation and must therefore satisfy cycle consistency. The objective function of enforcing this requirement can be expressed as:

$$\mathcal{L}_{\text{cycle}} = \mathop{\mathbb{E}}_{c, \mathbf{x}_{\epsilon(c)}} [d(f_\theta(f_\theta(\mathbf{x}_{\epsilon(c)}, \epsilon(c), c'), \epsilon(c'), c), \mathbf{x}_{\epsilon(c)})]. \quad (13)$$

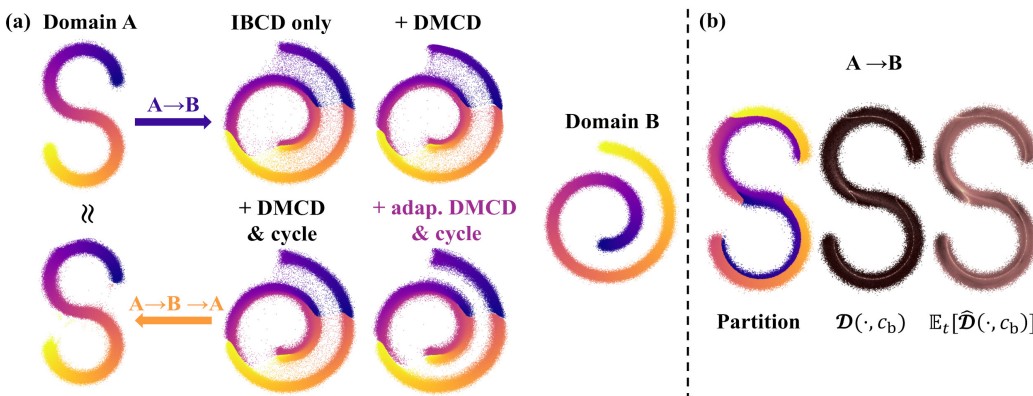

Figure 3: (a) Bidirectional translation results on a toy dataset which highlights each component's cumulative contributions. (b) Visualization of distillation difficulty $\mathcal{D}(\cdot, c_b)$ and its one-step approximation version $\mathbb{E}_t[\hat{\mathcal{D}}(\cdot, c_b)]$ of A→B translation. The function $g$ of $\hat{\mathcal{D}}$ was selected as a logarithm.

**Final Loss Functions.** The final loss, weighted by $\lambda_{\mathrm{DMCD}}$, $\lambda_{\mathrm{cycle}}$, for training $f_\theta$ is given by:

$$\theta^* = \arg\min_\theta \mathcal{L}_{\mathrm{IBCD}} + \lambda_{\mathrm{DMCD}}\mathcal{L}_{\mathrm{DMCD}} + \lambda_{\mathrm{cycle}}\mathcal{L}_{\mathrm{cycle}}. \tag{14}$$

Empirically, we found that the following adaptive training strategy further improves the performing: the training process begins with only the IBCD loss; as the student model approaches convergence, the DMCD and cycle consistency losses are additionally introduced to further refine the model's performance. Detailed training procedures and the complete algorithm can be found in Algorithm 2.

## 4 EXPERIMENTS

### 4.1 TOY DATA EXPERIMENT

To demonstrate the effectiveness of our framework in a controlled setting, we conducted bidirectional translation experiments using a two-dimensional synthetic toy dataset, where the two domains, $A$ and $B$, were selected as the S-curve and Swiss roll distributions, respectively.

**Validity of the IBCD.** Figure 3(a) shows the translation results from domain A→B for various models, highlighting the cumulative effectiveness of each component of our framework. Distillation using only the IBCD loss achieves basic translation, but some points are incorrectly mapped to low-density regions of the target domain. These points originate from the decision boundaries of the source domain (Appendix D.1). The addition of the DMCD loss improves translation by guiding more points toward high-density regions of the target domain. However, it does not effectively reposition target points that reside in low-density areas and instead reduces mode coverage by pushing points already in high-density regions to even denser areas. Introducing a cycle loss effectively alleviates the reduction in mode coverage caused by the introduction of DMCD and sharpens the decision boundaries within the target domain. Finally, incorporating the distillation difficulty adaptive weighting into DMCD selectively corrects points that have drifted into low-density regions, moving them toward higher-density areas. The complete cycle translation between domains (A→B→A) using a single model trained with our final approach effectively demonstrates the cycle consistency property, validating the robustness and fidelity of our method.

**Distillation Difficulty.** Figure 3(b) illustrates the impact of distillation difficulty on the translation process. On the left, we show the decision boundary of the source domain resulting from the translation from the target to the source domain by the DDIB teacher model. The middle and right panels depict $\mathcal{D}([x_{t_{-N+1}}, \cdots, x_{t_{N-1}}], c_b)$ and its expected one-step approximation, $\mathbb{E}_{t\sim\mathcal{U}[-N+1,N-2]}[\hat{\mathcal{D}}(x_t, c_b)]$ for the A→B translation, plotted at the source domain location $x_{\epsilon(c_a)}$. The distillation difficulty measure effectively captures the decision boundary, indicating challenging regions for the student model. As shown, its one-step approximation provides an accurate and

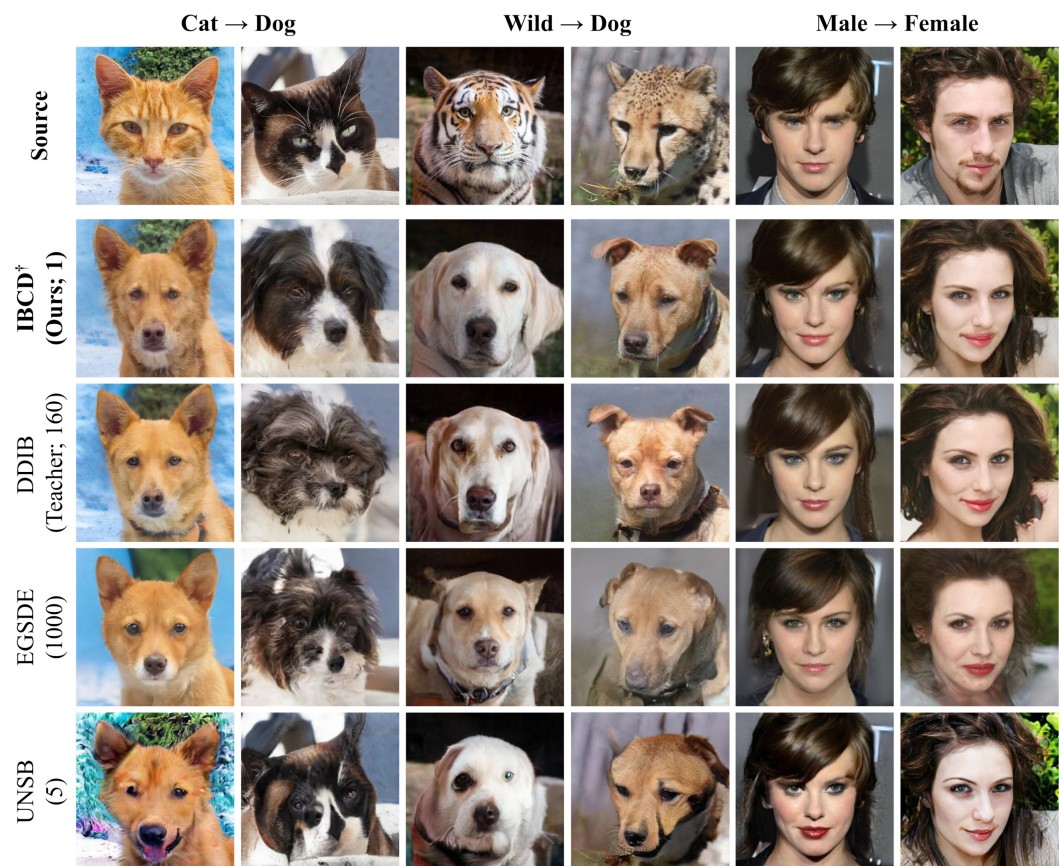

Figure 4: **Qualitative comparison of unpaired image-to-image translation tasks**. Compared to other diffusion-based baselines, our model achieves more realistic and source-faithful translations in a single step. The numbers in parentheses represent inference NFE.

suitable representation of the distillation difficulty, demonstrating its utility in guiding the training process and improving translation accuracy.

## 4.2 UNPAIRED IMAGE-TO-IMAGE TRANSLATION

In this section, we apply IBCD to various image-to-image (I2I) translation tasks, which are our primary focus. We then perform a comprehensive evaluation of the method's performance across these tasks to demonstrate its effectiveness and robustness.

**Evaluation.** Following the evaluation methodology and code from EGSDE (Zhao et al., 2022), a widely used benchmark for unpaired I2I tasks, we evaluated our approach on the Cat→Dog, Wild→Dog tasks from the AFHQ dataset (Choi et al., 2020) and the Male→Female tasks from the CelebA-HQ dataset (Karras, 2018). Initially, we trained AFHQ EDM and CelebA-HQ EDM models to serve as teacher models. One-step Cat↔Dog and Wild↔Dog translation models were distilled from the AFHQ EDM, while Male↔Female translation model was distilled from the CelebA-HQ EDM. For training and evaluation, all datasets were resized to 256 pixels. The evaluation metrics used were Fréchet Inception Distance (FID) (Heusel et al., 2017) and Density-Coverage (Naeem et al., 2020) to assess the realism of the translation, and PSNR and SSIM (Wang et al., 2004) to evaluate the faithfulness of the translation with the original images.

**Baselines.** As baselines, we compare our method against several GAN-based methods, including CycleGAN (Zhu et al., 2017), Self-Distance (Benaim & Wolf, 2017), GcGAN (Fu et al., 2019), LeSeSIM (Zheng et al., 2021), StarGAN v2 (Choi et al., 2020), and CUT (Park et al., 2020). We also benchmark against diffusion model (DM)-based methods such as ILVR (Choi et al., 2021),

Table 2: **Quantitative comparison of unpaired image-to-image translation tasks**. Most results are from the EGSDE paper, except those marked with *, which are from our re-implementation. We additionally measured the density-coverage metric (Naeem et al., 2020). Marker † indicates a hyperparameter configuration prioritizes reality over faithfulness.

| Method | NFE | FID↓ | PSNR ↑ | SSIM ↑ | Density ↑ | Coverage ↑ |
|---|---|---|---|---|---|---|
| **Cat→Dog** | | | | | | |
| CycleGAN (Zhu et al., 2017) | 1 | 85.9 | - | - | - | - |
| Self-Distance (Benaim & Wolf, 2017) | 1 | 144.4 | - | - | - | - |
| GcGAN (Fu et al., 2019) | 1 | 96.6 | - | - | - | - |
| LeSeSIM (Zheng et al., 2021) | 1 | 72.8 | - | - | - | - |
| StarGAN v2 (Choi et al., 2020) | 1 | $54.88 \pm 1.01$ | $10.63 \pm 0.10$ | $0.270 \pm 0.003$ | - | - |
| CUT (Park et al., 2020) | 1 | 76.21 | 17.48 | 0.601 | 0.971 | 0.696 |
| UNSB* (Kim et al., 2024a) | 5 | 68.59 | 17.65 | 0.587 | 1.045 | 0.706 |
| ILVR (Choi et al., 2021) | 1000 | $74.37 \pm 1.55$ | $17.77 \pm 0.02$ | $0.363 \pm 0.001$ | 1.036 | 0.572 |
| SDEdit (Meng et al., 2022) | 1000 | $74.17 \pm 1.01$ | $19.19 \pm 0.01$ | $0.423 \pm 0.001$ | 0.996 | 0.524 |
| EGSDE (Zhao et al., 2022) | 1000 | $65.82 \pm 0.77$ | $19.31 \pm 0.02$ | $0.415 \pm 0.001$ | 1.253 | 0.664 |
| EGSDE† (Zhao et al., 2022) | 1200 | $51.04 \pm 0.37$ | $17.17 \pm 0.02$ | $0.361 \pm 0.001$ | 1.540 | 0.836 |
| CycleDiffusion (Wu & De la Torre, 2023) | 1000(+100) | $58.87 \pm (-)$ | $18.50 \pm (-)$ | $0.557 \pm (-)$ | 0.894 | 0.786 |
| SDDM (Sun et al., 2023) | 100 | $62.29 \pm 0.63$ | - | $0.422 \pm 0.001$ | - | - |
| SDDM† (Sun et al., 2023) | 120 | $49.43 \pm 0.23$ | - | $0.361 \pm 0.001$ | - | - |
| DDIB* (Teacher) (Su et al., 2023) | 160 | 38.91 | 17.58 | 0.588 | 1.528 | 0.934 |
| **IBCD (Ours)** | 1 | **47.42** | **19.50** | **0.701** | 1.416 | **0.938** |
| **IBCD† (Ours)** | 1 | **44.69** | 18.04 | **0.663** | 1.542 | 0.934 |
| **Wild→Dog** | | | | | | |
| CUT (Park et al., 2020) | 1 | 92.94 | 17.20 | 0.592 | - | - |
| UNSB* (Kim et al., 2024a) | 5 | 70.03 | 16.86 | 0.573 | 1.035 | 0.704 |
| ILVR (Choi et al., 2021) | 1000 | $75.33 \pm 1.22$ | $16.85 \pm 0.02$ | $0.287 \pm 0.001$ | 1.313 | 0.548 |
| SDEdit (Meng et al., 2022) | 1000 | $68.51 \pm 0.65$ | $17.98 \pm 0.01$ | $0.343 \pm 0.001$ | 1.270 | 0.620 |
| EGSDE (Zhao et al., 2022) | 1000 | $59.75 \pm 0.62$ | $18.14 \pm 0.02$ | $0.343 \pm 0.001$ | 1.473 | 0.668 |
| EGSDE† (Zhao et al., 2022) | 1200 | $50.43 \pm 0.52$ | $16.40 \pm 0.01$ | $0.300 \pm 0.001$ | **1.714** | 0.776 |
| CycleDiffusion (Wu & De la Torre, 2023) | 1000(+100) | $56.45 \pm (-)$ | $17.82 \pm (-)$ | $0.479 \pm (-)$ | 1.013 | 0.814 |
| SDDM (Sun et al., 2023) | 100 | $57.38 \pm 0.53$ | - | $0.328 \pm 0.001$ | - | - |
| DDIB* (Teacher) (Su et al., 2023) | 160 | 38.59 | 17.03 | 0.552 | 1.594 | 0.924 |
| **IBCD (Ours)** | 1 | **48.68** | **18.25** | **0.653** | 1.541 | **0.920** |
| **IBCD† (Ours)** | 1 | **46.10** | 16.78 | **0.612** | 1.579 | 0.918 |
| **Male→Female** | | | | | | |
| CUT (Park et al., 2020) | 1 | 31.94 | 19.87 | 0.74 | - | - |
| UNSB* (Kim et al., 2024a) | 5 | 28.62 | 19.55 | 0.687 | 0.576 | 0.635 |
| ILVR (Choi et al., 2021) | 1000 | $46.12 \pm 0.33$ | $18.59 \pm 0.02$ | $0.510 \pm 0.001$ | - | - |
| SDEdit (Meng et al., 2022) | 1000 | $49.43 \pm 0.47$ | $20.03 \pm 0.01$ | $0.572 \pm 0.000$ | 0.788 | 0.373 |
| EGSDE (Zhao et al., 2022) | 1000 | $41.93 \pm 0.11$ | $20.35 \pm 0.01$ | $0.574 \pm 0.000$ | 0.880 | 0.453 |
| EGSDE† (Zhao et al., 2022) | 1200 | $30.61 \pm 0.19$ | $18.32 \pm 0.02$ | $0.510 \pm 0.001$ | 0.966 | 0.657 |
| SDDM (Sun et al., 2023) | 100 | $44.37 \pm 0.23$ | - | $0.526 \pm 0.001$ | - | - |
| DDIB* (Teacher) (Su et al., 2023) | 160 | 23.69 | 18.70 | 0.664 | 0.969 | 0.808 |
| **IBCD (Ours)** | 1 | **24.89** | **20.51** | **0.749** | **1.150** | **0.811** |
| **IBCD† (Ours)** | 1 | **24.71** | 20.11 | **0.744** | 1.144 | 0.801 |

SDEdit (Meng et al., 2022), EGSDE (Zhao et al., 2022), CycleDiffusion (Wu & De la Torre, 2023), and SDDM (Sun et al., 2023). Additionally, we compare our approach with UNSB (Kim et al., 2024a), a few-step Schrödinger bridge-based method, and the teacher DDIB (Su et al., 2023). Most of the comparison results are sourced from the EGSDE paper, while the results for UNSB and DDIB are based on our re-implementations.

**Comparison results.** Figure 4 and Table 2 present qualitative and quantitative comparison results between IBCD and baselines. The hyperparameter configuration for IBCD emphasizes a balance between faithfulness and realism, while the configuration for IBCD† prioritizes realism. Our framework consistently outperforms the baselines across various tasks and metrics, demonstrating the effectiveness of its components in improving the trade-off between faithfulness and reality. Although the student model exhibits a decrease in realism compared to the teacher, it shows enhanced faithfulness. This reduction in realism may be attributed to errors from the distillation process, the one-step conversion, and other contributing factors. Unlike the teacher, the student model simultaneously incorporates information about both domains, which may cause it to prioritize faithfulness along the trade-off curve. In some instances, the student's samples even surpass the teacher in terms of realism, potentially due to the additional training dynamics introduced by auxiliary losses beyond the

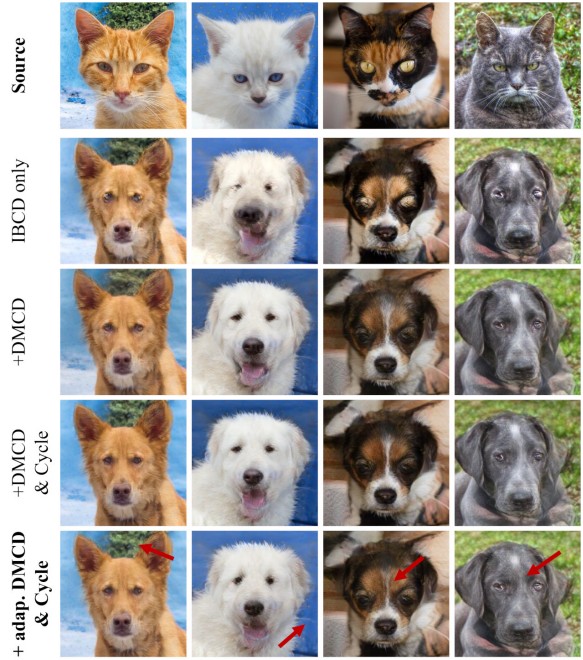

Figure 5: Qualitative ablation study results of IBCD in the Cat→Dog task.

Table 3: Quantitative ablation study results in the Cat→Dog task under the lowest FID.

| Component | FID↓ | PSNR ↑ Teacher | Source |
|---|---|---|---|
| IBCD only | 48.12 | 18.27 | 19.02 |
| + DMCD | 44.40 | 17.95 | 16.80 |
| + DMCD & Cycle | 44.31 | 18.22 | 17.19 |
| **+ adap. DMCD & Cycle** | 44.69 | 18.97 | 18.04 |

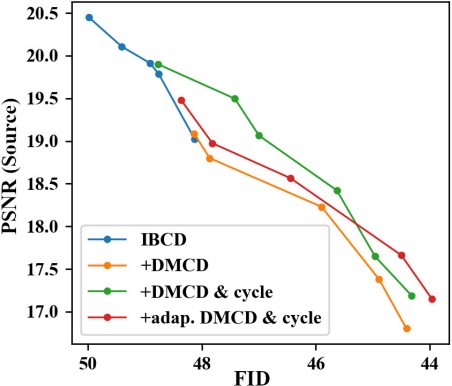

Figure 6: Ablation study results demonstrating improved PSNR-FID trade-off for the Cat→Dog task.

IBCD loss. This suggests that the student's ability to leverage information from both domains and auxiliary training components can lead to further refinement and improvement in performance.

**Ablation Study.** We conducted an ablation study on the Cat→Dog task to assess component effectiveness. For the ablation study, DMCD loss, cycle loss, and distillation difficulty adaptive weighting (*i.e.* adaptive DMCD) were sequentially added to the IBCD loss-only model. Additionally, to measure distillation error, we calculated PSNR relative to the DDIB teacher results (PSNR-teacher), complementing the standard PSNR used in the Table 2 (PSNR-source). Figure 5 and Table 3 show results for each ablated model that achieved the lowest FID. Figure 6 presents PSNR-FID trade-off curves for various hyperparameters ($\lambda_{\text{IBCD}}$, $\lambda_{\text{cycle}}$, and training steps) for each ablated model. Each added component leads to a significant reduction in FID beyond the lower bound achievable by vanilla IBCD, while minimizing the PSNR degradation due to the inherent trade-off of the task, and minimizing distillation error. In particular, adaptive DMCD is effective when the lowest FID is desired in the trade-off curve. These results confirm that the components of IBCD collectively contribute to the improvement of the tradeoff between faithfulness and reality.

## 5 CONCLUSION

In this work, we introduced a novel unpaired bidirectional one-step image translation framework, Implicit Bridge Consistent Distillation (IBCD). By distilling the diffusion implicit bridge through an extended consistency distillation framework, we achieved bidirectional translation without the need for paired data or adversarial training. Our approach addresses the limitations of traditional consistency distillation through the proposed Distribution Matching for Consistency Distillation (DMCD) and distillation difficulty adaptive weighting strategies. Empirical evaluations on both toy and high-dimensional datasets demonstrate the effectiveness and scalability of IBCD. We believe that IBCD represents a significant advancement in the field of general one-step image translation, providing a versatile and efficient solution for various image tasks, including image restoration, especially in scenarios with limited paired data.

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

# SUPPLEMENTARY MATERIAL

## A    ALGORITHMS

In this section, we present the vanilla implicit bridge consistency distillation algorithm (Algorithm 1), which utilizes only the IBCD losses. Additionally, we introduce the final implicit bridge consistency distillation algorithm (Algorithm 2), which incorporates all the losses discussed in the text, including DMCD and adaptive weighting strategies, to enhance performance and address the limitations identified in the vanilla version.

---

**Algorithm 1:** (Vanilla) Implicit Bridge Consistent Distillation (IBCD)

---

**Input:** Teacher diffusion model $\phi$, datasets $\mathcal{S}_A$ and $\mathcal{S}_B$, class conditions $c_a$ and $c_b$.

1  $j \leftarrow 0, \theta \leftarrow \phi, \theta^- \leftarrow \phi$

2  **repeat**

3      $c \leftarrow$ **if** $(j\%2 == 0$ **then** $c_a$ **else** $c_b)$

4      Sample $\mathbf{x}^a \sim \mathcal{S}_A, \mathbf{x}^b \sim \mathcal{S}_B$

5      **if** $c == c_b$ **then**

6          Sample $n_a \sim \mathcal{U}[-N+1, -1], n_b \sim \mathcal{U}[0, N-2]$

7      **else**

8          Sample $n_a \sim \mathcal{U}[-N+2, 0], n_b \sim \mathcal{U}[1, N-1]$

9      Sample $\mathbf{x}_{t_{n_a}} \sim \mathcal{N}(\mathbf{x}^a, t_{n_a}^2 \mathbf{I}), \mathbf{x}_{t_{n_b}} \sim \mathcal{N}(\mathbf{x}^b, t_{n_b}^2 \mathbf{I})$

10      **if** $c == c_b$ **then**

11          Estimate $\hat{\mathbf{x}}_{t_{n_a+1}}, \hat{\mathbf{x}}_{t_{n_b+1}}$ with Eq. (8)

12      **else**

13          Estimate $\hat{\mathbf{x}}_{t_{n_a-1}}, \hat{\mathbf{x}}_{t_{n_b-1}}$ with Eq. (9)

14      $\mathbf{t}_1 \leftarrow [t_{n_a}; t_{n_b}], \mathbf{t}_2 = [t_{n_a \pm 1}; t_{n_b \pm 1}]$

15      $\mathbf{x}_{\mathbf{t}_1} \leftarrow [\mathbf{x}_{t_{n_a}}; \mathbf{x}_{t_{n_b}}], \hat{\mathbf{x}}_{\mathbf{t}_2} \leftarrow [\hat{\mathbf{x}}_{t_{n_a \pm 1}}; \hat{\mathbf{x}}_{t_{n_b \pm 1}}]$

16      $\mathcal{L}_{\text{IBCD}} \leftarrow [\lambda(\mathbf{t}_2) d_{\text{IBCD}}(f_\theta(\mathbf{x}_{\mathbf{t}_1}, \mathbf{t}_1, c), f_{\theta^-}(\hat{\mathbf{x}}_{\mathbf{t}_2}, \mathbf{t}_2, c))]$

17      $\theta \leftarrow \theta - \zeta_\theta \nabla_\theta \mathcal{L}_{\text{IBCD}}$

18      $\theta^- \leftarrow \text{sg}(\mu\theta^- + (1-\mu)\theta)$

19      $j \leftarrow j+1$

20  **until** $\mathcal{L}_{IBCD}$ *convergence*;

    **Output:** Unified one-step model $f_\theta$ for bidirectional image translation.

---

---

**Algorithm 2:** (Final) Implicit Bridge Consistent Distillation (IBCD)

---

**Input:** Teacher diffusion model $\phi$, datasets $\mathcal{S}_A$ and $\mathcal{S}_B$, class conditions $c_a$ and $c_b$.

1 $j \leftarrow 0, \theta \leftarrow \phi, \theta^- \leftarrow \phi, \psi \leftarrow \phi$

2 **repeat**

3    $c \leftarrow$ **if** $(j\%2 == 0$ **then** $c_a$ **else** $c_b)$

4    Sample $\mathbf{x}^a \sim \mathcal{S}_A,\ \mathbf{x}^b \sim \mathcal{S}_B$

   //

   // IBCD loss

5    **if** $c == c_b$ **then**

6       $\lfloor$ Sample $n_a \sim \mathcal{U}[-N+1, -1],\ n_b \sim \mathcal{U}[0, N-2]$

7    **else**

8       $\lfloor$ Sample $n_a \sim \mathcal{U}[-N+2, 0],\ n_b \sim \mathcal{U}[1, N-1]$

9    Sample $\mathbf{x}_{t_{n_a}} \sim \mathcal{N}(\mathbf{x}^a, t_{n_a}^2\mathbf{I}),\ \mathbf{x}_{t_{n_b}} \sim \mathcal{N}(\mathbf{x}^b, t_{n_b}^2\mathbf{I})$

10    **if** $c == c_b$ **then**

11       $\lfloor$ Estimate $\hat{\mathbf{x}}_{t_{n_a}+1},\ \hat{\mathbf{x}}_{t_{n_b}+1}$ with Eq. (8)

12    **else**

13       $\lfloor$ Estimate $\hat{\mathbf{x}}_{t_{n_a}-1},\ \hat{\mathbf{x}}_{t_{n_b}-1}$ with Eq. (9)

14    $\mathbf{t}_1 \leftarrow [t_{n_a}; t_{n_b}],\ \mathbf{t}_2 = [t_{n_a\pm1}; t_{n_b\pm1}]$

15    $\mathbf{x}_{\mathbf{t}_1} \leftarrow [\mathbf{x}_{t_{n_a}}; \mathbf{x}_{t_{n_b}}],\ \hat{\mathbf{x}}_{\mathbf{t}_2} \leftarrow [\hat{\mathbf{x}}_{t_{n_a\pm1}}; \hat{\mathbf{x}}_{t_{n_b\pm1}}]$

16    $\mathcal{L}_{\text{IBCD}} \leftarrow [\lambda(\mathbf{t}_2)d_{\text{IBCD}}(f_\theta(\mathbf{x}_{\mathbf{t}_1}, \mathbf{t}_1, c), f_{\theta^-}(\hat{\mathbf{x}}_{\mathbf{t}_2}, \mathbf{t}_2, c))]$

   //

   // DMCD loss

17    Sample $i \sim \mathcal{U}[0, N-1]$

18    Sample $\mathbf{x}_{t_i} \sim \mathcal{N}(f_\theta(\mathbf{x}_{\mathbf{t}_1}, \mathbf{t}_1, c), t_i^2\mathbf{I})$

19    $\hat{\mathcal{D}} \leftarrow \text{sg}(g(d_{\text{DMCD}}(f_\theta(\mathbf{x}_{\mathbf{t}_1}, \mathbf{t}_1, c), f_{\theta^-}(\hat{\mathbf{x}}_{\mathbf{t}_2}, \mathbf{t}_2, c))))$

20    $\nabla_\theta \mathcal{L}_{\text{DMCD}} \leftarrow w_{t_i}\hat{\mathcal{D}} \cdot (s_\psi(\mathbf{x}_{t_i}, t_i, c) - s_\phi(\mathbf{x}_{t_i}, t_i, c))\nabla_\theta f_\theta(\mathbf{x}_{\mathbf{t}_1}, \mathbf{t}_1, c)$

   //

   // Cycle loss

21    Sample $\mathbf{x}_{\epsilon(c_a)} \sim \mathcal{N}(\mathbf{x}^a, \sigma_{\min}^2\mathbf{I}),\ \mathbf{x}_{\epsilon(c_b)} \sim \mathcal{N}(\mathbf{x}^b, \sigma_{\min}^2\mathbf{I})$

22    $\mathbf{t}_3 \leftarrow [\epsilon(c_a); \epsilon(c_b)],\ \mathbf{t}_4 \leftarrow [\epsilon(c_b); \epsilon(c_a)]$

23    $\mathbf{c}_3 \leftarrow [c_b; c_a],\ \mathbf{c}_4 \leftarrow [c_a; c_b]$

24    $\mathbf{x}_{\mathbf{t}_3} \leftarrow [\mathbf{x}_{\epsilon(c_a)}; \mathbf{x}_{\epsilon(c_b)}]$

25    $\mathcal{L}_{\text{cycle}} \leftarrow d_{\text{cycle}}(f_\theta(f_\theta(\mathbf{x}_{\mathbf{t}_3}, \mathbf{t}_3, \mathbf{c}_3), \mathbf{t}_4, \mathbf{c}_4), \mathbf{x}_{\mathbf{t}_3})$

   //

   // Optimize the student

26    $\nabla_\theta \mathcal{L}_{\text{total}} \leftarrow \nabla_\theta \mathcal{L}_{\text{IBCD}} + \lambda_{\text{DMCD}}\nabla_\theta \mathcal{L}_{\text{DMCD}} + \lambda_{\text{cycle}}\nabla_\theta \mathcal{L}_{\text{cycle}}$

27    $\theta \leftarrow \theta - \zeta_\theta \nabla_\theta \mathcal{L}_{\text{total}}$

28    $\theta^- \leftarrow \text{sg}(\mu\theta^- + (1-\mu)\theta)$

   //

   // Optimize the fake DM

29    $\mathcal{L}_{DSM} \leftarrow$ DSM loss of EDM with sample $f_\theta(\mathbf{x}_{\mathbf{t}_1}, \mathbf{t}_1, c)$, class condition $c$, and fake DM $\phi$

30    $\phi \leftarrow \phi - \zeta_\phi \nabla_\phi \mathcal{L}_{\text{DSM}}$

31    $j \leftarrow j + 1$

32 **until** $\mathcal{L}_{total}$ *convergence*;

**Output:** Unified one-step model $f_\theta$ for bidirectional image translation.

---

## B  EXTENDING EDM/CD FOR THE IBCD

The EDM (Karras et al., 2022) parametrization for the student $f_\theta$ in consistency distillation (Song et al., 2023) is defined as follows for positive real-valued $t$ and the neural network $F_\theta$:

$$f_\theta(\mathbf{x}_t, t) = c_{\text{skip}}(t)\mathbf{x}_t + c_{\text{out}}(t)F_\theta(c_{\text{in}}(t)\mathbf{x}_t, t'(t)). \tag{15}$$

In CD, authors choose

$$c_{\text{skip}}(t) = \frac{\sigma_{\text{data}}^2}{(t-\epsilon)^2 + \sigma_{\text{data}}^2}, \qquad c_{\text{out}}(t) = \frac{\sigma_{\text{data}}(t-\epsilon)}{\sqrt{\sigma_{\text{data}}^2 + t^2}}, \qquad c_{\text{in}}(t) = \frac{1}{\sqrt{\sigma_{\text{data}}^2 + t^2}}, \tag{16}$$

$$t'(t) = 250 \cdot \ln(t + 10^{-44}) \tag{17}$$

to satisfies the boundary condition $f(\mathbf{x}_\epsilon, \epsilon) = \mathbf{x}_\epsilon$, and rescales the timestep.

For IBCD, we parametrize the student $f_\theta$ for non-zero real-valued $t$ and target domain condition $c$ as:

$$f_\theta(\mathbf{x}_t, t, c) = c_{\text{skip}}(t, c)\mathbf{x}_t + c_{\text{out}}(t, c)F_\theta(c_{\text{in}}(t, c)\mathbf{x}_t, t'(t)), \tag{18}$$

which reflects the necessity for $c_{\text{skip}}, c_{\text{out}}$, and $c_{\text{in}}$ depend on target domain condition $c$, ensuring that the proper boundary conditions can be applied at $t = \epsilon(c)$ depending on the target domain $c \in \{c_a, c_b\}$ direction.

Although the student model is fully trained during the distillation process and does not theoretically need to be compatible with the teacher model, initializing it using the teacher model makes it advantageous to design the student to be as compatible as possible. We select $c_{\text{skip}}, c_{\text{out}}$, and $c_{\text{in}}$ according to Eq. (19), (20), (21), ensuring continuity and compliance with the new boundary conditions while maintaining the definitions within the target domain regions ($t > 0$ for $c = c_b$, $t < 0$ for $c = c_a$).

$$c_{\text{skip}}(t, c) = \begin{cases} \frac{1+\text{sign}(t)}{2} \frac{\sigma_{\text{data}}^2}{(t-\epsilon(c))^2 + \sigma_{\text{data}}^2} & \text{if } c = c_b \\ \frac{1+\text{sign}(-t)}{2} \frac{\sigma_{\text{data}}^2}{(t-\epsilon(c))^2 + \sigma_{\text{data}}^2} & \text{if } c = c_a \end{cases} \tag{19}$$

$$c_{\text{out}}(t, c) = \begin{cases} \frac{1+\text{sign}(t)}{2} \frac{\sigma_{\text{data}}(t-\epsilon(c))}{\sqrt{\sigma_{\text{data}}^2 + t^2}} + \frac{1-\text{sign}(t)}{2}\sigma_{\text{data}} & \text{if } c = c_b \\ -\frac{1+\text{sign}(-t)}{2} \frac{\sigma_{\text{data}}(t-\epsilon(c))}{\sqrt{\sigma_{\text{data}}^2 + t^2}} + \frac{1-\text{sign}(-t)}{2}\sigma_{\text{data}} & \text{if } c = c_a \end{cases} \tag{20}$$

$$c_{\text{in}}(t, c) = \frac{1}{\sqrt{\sigma_{\text{data}}^2 + t^2}} \tag{21}$$

We also extend the timestep rescaler as Eq. (22) to a symmetric and continuous form, ensuring shape compatibility with the original positive-bound domain. This symmetric design reflects the fact that the sign of the timestep separates the domains, while its absolute value represents the noise magnitude:

$$t'(t) = 250 \cdot \text{sign}(t)(\ln(|t| + 10^{-3}) - \ln(\sigma_{\max} + 10^{-44})). \tag{22}$$

This approach preserves the structural integrity of the model and maintains consistent behavior across both domains. The parametrization extension of EDM/CD, as presented here, is visually illustrated in Figure 7.

## C  IMPLEMENTATION DETAILS

**Model Architectures.** All models used in this study – the teacher $\phi$, student $\theta$, and fake DM $\psi$ – employed the same model architecture as in EDM/CD (Karras et al., 2022; Song et al., 2023). The architecture configuration followed that of the LSUN-256 teacher EDM model introduced by Song et al. (2023). However, the student model was further modified with the model parametrization described in Appendix B, while the teacher and fake DM maintained the original EDM parametrization.

**Teacher Model Training.** The teacher model was trained using the EDM implementation and the LSUN-256 model training configuration provided by Song et al. (2023). The training setup included a log-normal schedule sampler and L2 loss, with a global batch size of 288, a learning rate of 1e-4, a dropout rate of 0.1, and an exponential moving average (EMA) of 0.9999. Mixed precision training

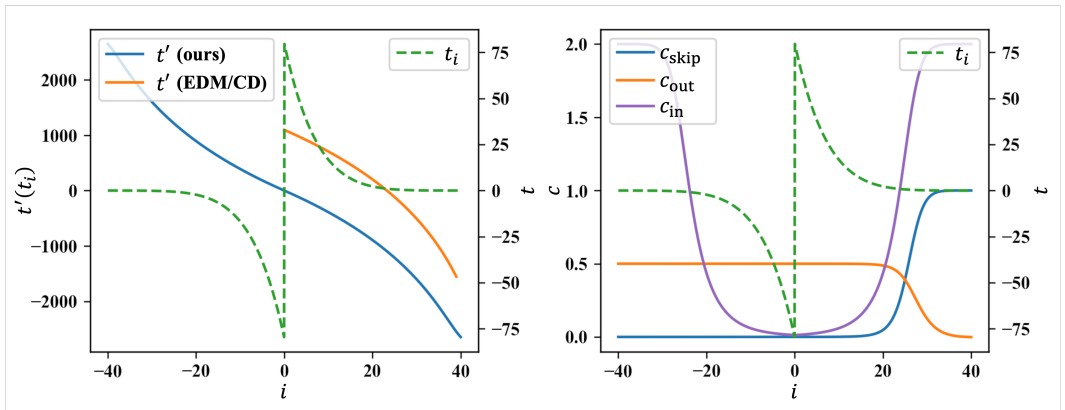

Figure 7: **Extension of EDM/CD model formulation for negative $t$ in IBCD student model.** $c_{\text{skip}}, c_{\text{out}}$, and $c_{\text{in}}$ represent when $c = c_b$ (the translation direction is $\mathcal{X}_A \to \mathcal{X}_B$).

was enabled, and weight decay was not applied. The teacher model was trained with class conditions on two types of AFHQ-256 models (cat, dog, and wild) and CelebA-HQ-256 models (female and male). The AFHQ and CelebA-HQ models were trained using their respective training sets from the AFHQ (Choi et al., 2020) and CelebA-HQ (Karras, 2018) datasets. Each model was trained for approximately 5 days, completing 800K steps on an NVIDIA A100 40GB eight GPU setup.

**Implicit Bridge Consistency Distillation.** The discretization of DDIB trajectories is defined by extending the sampling discretization of EDM to satisfy Eq. (6):

$$t_i = \sigma_i = \begin{cases} \text{sign}(i)(\sigma_{\max}^{1/\rho} + \frac{|i|}{N-1}(\sigma_{\min}^{1/\rho} - \sigma_{\max}^{1/\rho}))^\rho & (N < i < N) \\ 0 & (i = \pm N) \end{cases} \tag{23}$$

where $\quad \text{sign}(x) = \begin{cases} +1 & (x \geq 0) \\ -1 & (x < 0) \end{cases}, \sigma_{\min} = 0.002, \ \sigma_{\max} = 80, \ \sigma_{\text{data}} = 0.5, \ N = 40, \ \rho = 7.0.$

For the distance function $d$ in each loss, $d_{\text{IBCD}}$ and $d_{\text{DMCD}}$ were based on LPIPS (Zhang et al., 2018), while $d_{\text{cycle}}$ used the L1 loss. The EMA parameter of the EMA model $\theta^-$ was 0.95, and an additional EMA with a separate parameter 0.9999432189950708 was applied to the student model $\theta$ and used during inference. The global batch size was 256, with the student learning rate of 4e-5 and the fake DM learning rate of 1e-4. Dropout and weight decay were not used, and mixed precision learning was employed.

The ODE solver used was the 2nd order Huen solver (Ascher & Petzold, 1998), consistent with EDM/CD. The weight scheduler for the IBCD loss employed $\lambda(t) = 1$, while the DMCD loss used the weight scheduler $w_t$ as suggested in Yin et al. (2024). For the three tasks, Cat↔Dog, Wild↔Dog models were distilled using the AFHQ-256 teacher model and its corresponding training dataset. The Male↔Female models were distilled using the CelebA-HQ-256 teacher model and its training dataset.

The distillation process began with only the IBCD loss and transitioned to using the full loss set once the FID (Heusel et al., 2017) evaluation metrics stabilized (*i.e.* transition step). Distillation was conducted on the same NVIDIA A100 40GB eight hardware used for training the teacher model. Additional hyperparameters for each model and configuration are detailed in Table 4.

**Evaluation.** We followed the evaluation methodology and tasks outlined in EGSDE (Zhao et al., 2022). The publicly available evaluation code[2] was used without modification. Validation sets from the AFHQ and CelebA-HQ datasets were used as the evaluation datasets. All images in each validation set were translated using the respective task-specific models. For each image pair (source domain and translated target domain), PSNR and SSIM were computed, and the average values across all pairs were reported.

---

[2]https://github.com/ML-GSAI/EGSDE

Table 4: Specific hyperparameters employed by different models and configurations.

| Model | Cat↔Dog | | Wild↔Dog | | Male↔Female | |
|---|---|---|---|---|---|---|
| Configuration | IBCD | IBCD$^\dagger$ | IBCD | IBCD$^\dagger$ | IBCD | IBCD$^\dagger$ |
| $\lambda_{\text{DMCD}}$ | 1 | 0.18 | 0.2 | 0.2 | 0.02 | 0.02 |
| $\lambda_{\text{cycle}}$ | 0.03 | 0.003 | 0.001 | 0.0003 | 0.00001 | 0.00003 |
| $g(\cdot)$ | 1 | | $\min(\log(\cdot) + 10)$ | | | |
| transition step | 200K | 200K | 200K | 200k | 500K | 500K |
| total distillation step | 210K | 230K | 210K | 230K | 510K | 520K |

FID (Heusel et al., 2017) was calculated using the `pytorch-fid`[3] library to measure the distance between the real target domain image distribution and the translated target image distribution. Following the methodology of Choi et al. (2020) and Zhao et al. (2022), images from the CelebA-HQ dataset were resized and normalized before FID calculation, while images for other tasks were evaluated without additional preprocessing. L2 distance measurement was not included in this evaluation.

Density-coverage (Naeem et al., 2020) was computed using `prdc-cli`[4] between the distribution of real target domain images and the distribution of images translated into the target domain, similar to the FID measurement. The measurement mode was `T4096` (features of the `fc2` layer of the ImageNet pre-trained VGG16 (Simonyan, 2014) model). The metric was computed for the entire dataset at once, without using mini-batches. Unlike FID, no specific transformation was applied for the CelebA-HQ dataset.

**Reproductions.** To evaluate our method, we replicated UNSB and DDIB, two approaches that have not been previously evaluated on our benchmark datasets. For UNSB, we used the publicly available official code for both training and inference, following the default configuration for the Horse→Zebra task and training the model for 400 epochs. During inference, we performed 5 steps. For DDIB, we implemented the method within our framework. Specifically, DDIB was executed by first solving the ODE backward from the source domain, then solving it forward again to the target domain using the EDM model trained for IBCD. The ODE solver was implemented in the same manner as the EDM sampler, utilizing the same sampling hyperparameters defined for EDM/IBCD. This setup ensured consistency in the evaluation and allowed for a direct comparison of performance across methods.

We also re-sampled the result from models (CUT, ILVR, SDEdit, EGSDE, CycleDiffusion) for which the density-coverage (Naeem et al., 2020) metric was not originally reported. The density-coverage metric was measured for these models using the method described above and included the results in Table 2. The target of measurement for density-coverage was limited to baseline models that met the following criteria: 1) Open-source code and checkpoints were available. 2) FID, PSNR, and SSIM values reported by the authors could be reproduced using the reported sampling strategy. This ensured that all metrics in Table 2 were measured on consistent samples.

# D  FURTHER EXPERIMENTAL RESULTS

## D.1  DISTILLATION ERROR IN VANILLA IBCD

Figure 8 illustrates the distillation error that arises when using only vanilla IBCD loss on the synthetic toy dataset. When generating samples from pure noise to domain $B$ (Figure 8 (a)) or translating samples from domain $A$ to domain $B$ (Figure 8 (b)) using only IBCD loss, the translated results often fall in the low-density region of the target distribution. These translated points primarily originate from the source domain decision boundary, which is the boundary separating the partition in the source domain that should be mapped to two different target domain modes. Translation errors are more pronounced in longer neural jump paths, such as those involved in translations ($i = -N + 1 \rightarrow N - 1$), compared to shorter paths in generation ($i = 0 \rightarrow N - 1$).

---

[3]https://github.com/mseitzer/pytorch-fid

[4]https://github.com/Mahmood-Hussain/generative-evaluation-prdc

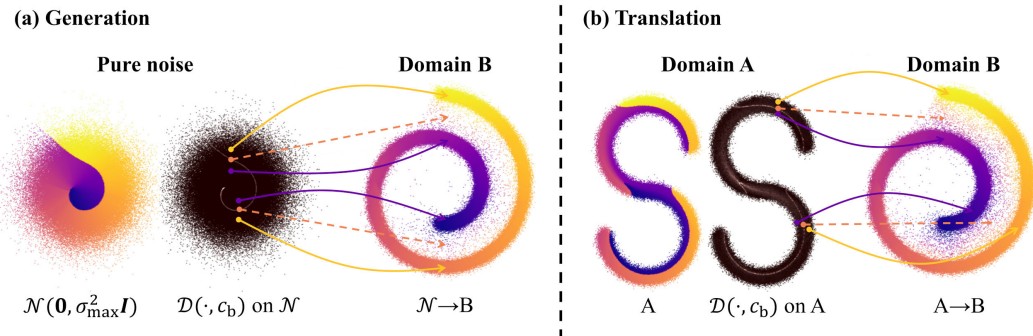

Figure 8: **Incorrect mapping to low-density regions due to the distillation error.** (a) Generation with vanilla IBCD and (b) translation with vanilla IBCD.

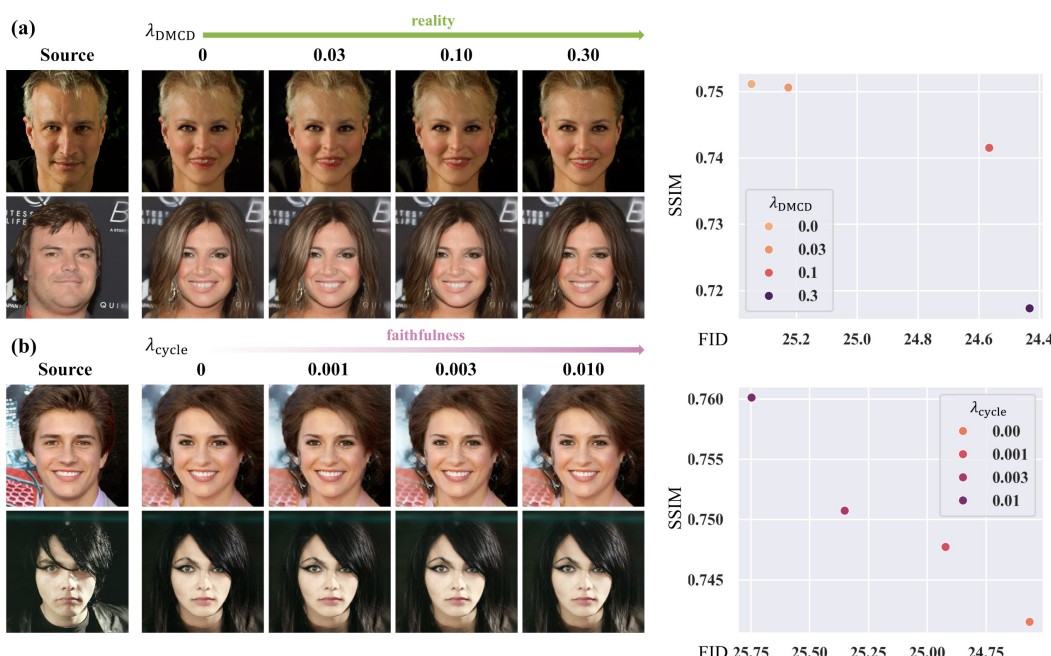

Figure 9: **Effect of the auxiliary loss weights ($\lambda_{\text{DMCD}}$, $\lambda_{\text{cycle}}$) for the Male→Female task.** In (a) $\lambda_{\text{cycle}}$ was set to 0, and in (b) $\lambda_{\text{DMCD}}$ was set to 0.10. Distillation difficulty adaptive waiting was not applied.

### D.2 Effect of the Auxiliary Loss Weights

Following the component ablation study of IBCD in the main text, we further investigated the influence of auxiliary loss weights on translation outcomes. Specifically, we varied the weight of the DMCD loss $\lambda_{\text{DMCD}}$ and the cycle loss $\lambda_{\text{cycle}}$ in the Male→Female task (Figure 9). During these experiments, distillation difficulty adaptive weighting was not applied. The results aligned with expectations: as $\lambda_{\text{DMCD}}$ increases, the realism of the translation result improved, while increasing $\lambda_{\text{cycle}}$ enhanced the faithfulness of the translation. Thus, in the realism-faithfulness trade-off curve, the DMCD loss emphasizes realism, whereas the cycle loss emphasizes faithfulness.

### D.3 Approximated Distillation Difficulty in Image-to-image Translation

To explore the implications of the approximated distillation difficulty for real image-to-image translation tasks, we computed an expected approximated distillation difficulty

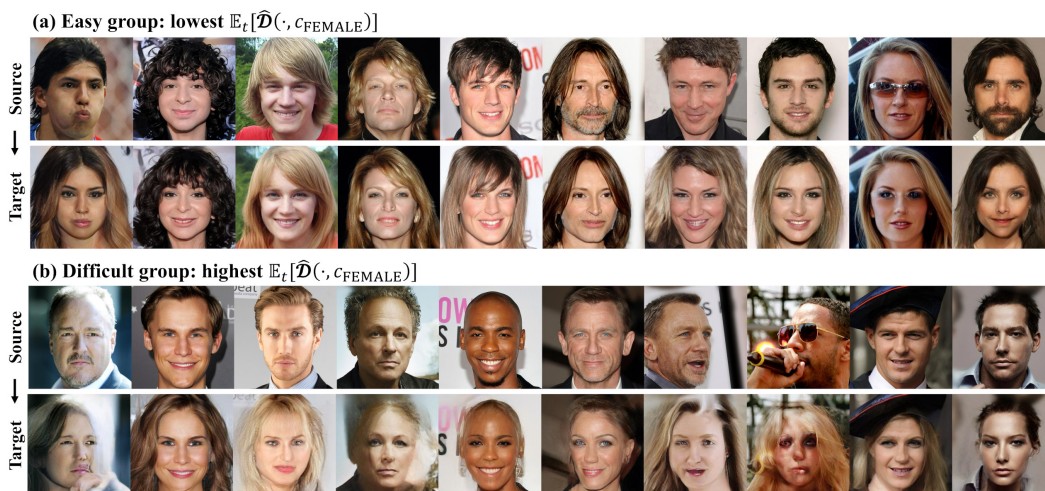

Figure 10: Relationship between self-assessed approximate distillation difficulty $\mathbb{E}_t[\hat{\mathcal{D}}(\cdot, c_{\text{FEMALE}})]$ and the translations performed in the Male→Female task.

Table 5: **Quantitative comparison of model inference times.** *Not supported parallel sampling.

| Method | Batch size | NFE | Time [$s/img$] ↓ | Relative Time ↓ |
|---|---|---|---|---|
| StarGAN v2 (Choi et al., 2020) | 256 | 1 | 0.058 | 5.5 |
| CUT (Park et al., 2020) | 1* | 1 | 0.068 | 6.4 |
| UNSB (Kim et al., 2024a) | 1* | 5 | 0.104 | 9.9 |
| ILVR (Choi et al., 2021) | 50 | 1000 | 12.915 | 1224.2 |
| SDEdit (Meng et al., 2022) | 70 | 1000 | 6.378 | 604.5 |
| EGSDE (Zhao et al., 2022) | 13 | 1000 | 15.385 | 1458.3 |
| CycleDiffusion (Wu & De la Torre, 2023) | 1* | 1000(+100) | 26.032 | 2467.5 |
| DDIB (Teacher) (Su et al., 2023) | 165 | 160 | 0.956 | 90.6 |
| **IBCD (Ours)** | 165 | 1 | **0.011** | **1** |

$\mathbb{E}_{t \sim \mathcal{U}[-N+1, N-2]}[\hat{\mathcal{D}}(\mathbf{x}_t, c_{\text{FEMALE}})]$ for all trajectories generated with the DDIB teacher in the Male→Female task using the vanilla IBCD model. We then selected the trajectories with the top 10 and bottom 10 approximate distillation difficulties and performed Male→Female translation using the vanilla IBCD model for these trajectories, as shown in Figure 10 without cherry-picking. The results indicate that the IBCD model struggles to effectively transform source images from trajectories with high approximate distillation difficulty into target images compared to those with low approximate distillation difficulty. Specifically, the translation results within the top 10 distillation difficulty group exhibit relatively inferior image quality, highlighting the impact of distillation difficulty on translation performance.

### D.4 MODEL INFERENCE EFFICIENCY

To reflect real-world constraints such as model size and inference algorithms, we conducted an inference speed comparison experiment. Instead of relying solely on NFE comparisons, we measured the actual inference time for a Cat→Dog task on a single NVIDIA GeForce RTX 4090 GPU. Table 5 presents the average inference time per image and the relative time for each methodology. The batch size was set to maximize GPU VRAM utilization (24 GB), and if the official code did not support parallel sampling, a batch size of 1 was used. The results demonstrate that our methodology is the most computationally efficient even in real-world sampling scenarios.

### D.5 FAILURE CASES

IBCD occasionally produces failure cases as illustrated in Figure 11. The primary failures can be attributed to incomplete translations (Figure 11(a)) and incorrect cycle translations (Figure 11(b)),

which are likely due to distillation errors and the side effects of auxiliary losses. Distillation errors from the CD, in particular, appear to be the primary reason. The DMCD and cycle translation loss can also contribute to these issues, with the former leading to incorrect cycle translations and the latter to incomplete translations. Minimizing distillation errors through improved distillation methods and advanced weighting strategies for auxiliary losses might address this issue.

### D.6    BIDIRECTIONAL TRANSLATIONS

To evaluate IBCD's bidirectional translation capabilities, we compared it to baseline methods through two tasks: *opposite translation* and *cycle translation*. Opposite translation involves reversing the main translation task (Dog→Cat, Dog→Wild, Female→Male), while cycle translation involves performing the reverse task after the main translation (Cat→Dog→Cat, Wild→Dog→Wild, Male→Female→Male). To ensure a fair comparison of bidirectional performance, we used the same model and sampling hyperparameters for each domain pair (Cat↔Dog, Wild↔Dog, Male↔Female) in both opposite and cycle translation tasks.

Given the limited number of models capable of bidirectional translation, we selected StarGAN v2 (Choi et al., 2020), CycleDiffusion (Wu & De la Torre, 2023), and DDIB (teacher) (Su et al., 2023) as baselines. We measured FID for the final target domain for the cycle translation task. It's worth noting that StarGAN v2's inference process differs from its main translation task (Table 2) performed by Zhao et al. (2022) for a better fair comparison. It inputs the same source image as both the source and reference images, enabling it to achieve both high reality and faithfulness.

Table 6 and Figures 12, 13 demonstrate that our model also excels in reverse and cycle translation tasks, exhibiting the best performance and high efficiency. This further supports its strong bidirectional translation capabilities.

### D.7    MORE QUALITATIVE RESULTS

In this section, we present additional qualitative results obtained through cycle translation tasks (Cat→Dog→Cat, Wild→Dog→Wild, Male→Female→Male). The results of the Cat↔Dog, Wild↔Dog, and Male↔Female model are illustrated in Figures 14, 15, 16. These results highlight our model's one-way and bidirectional translation capabilities.

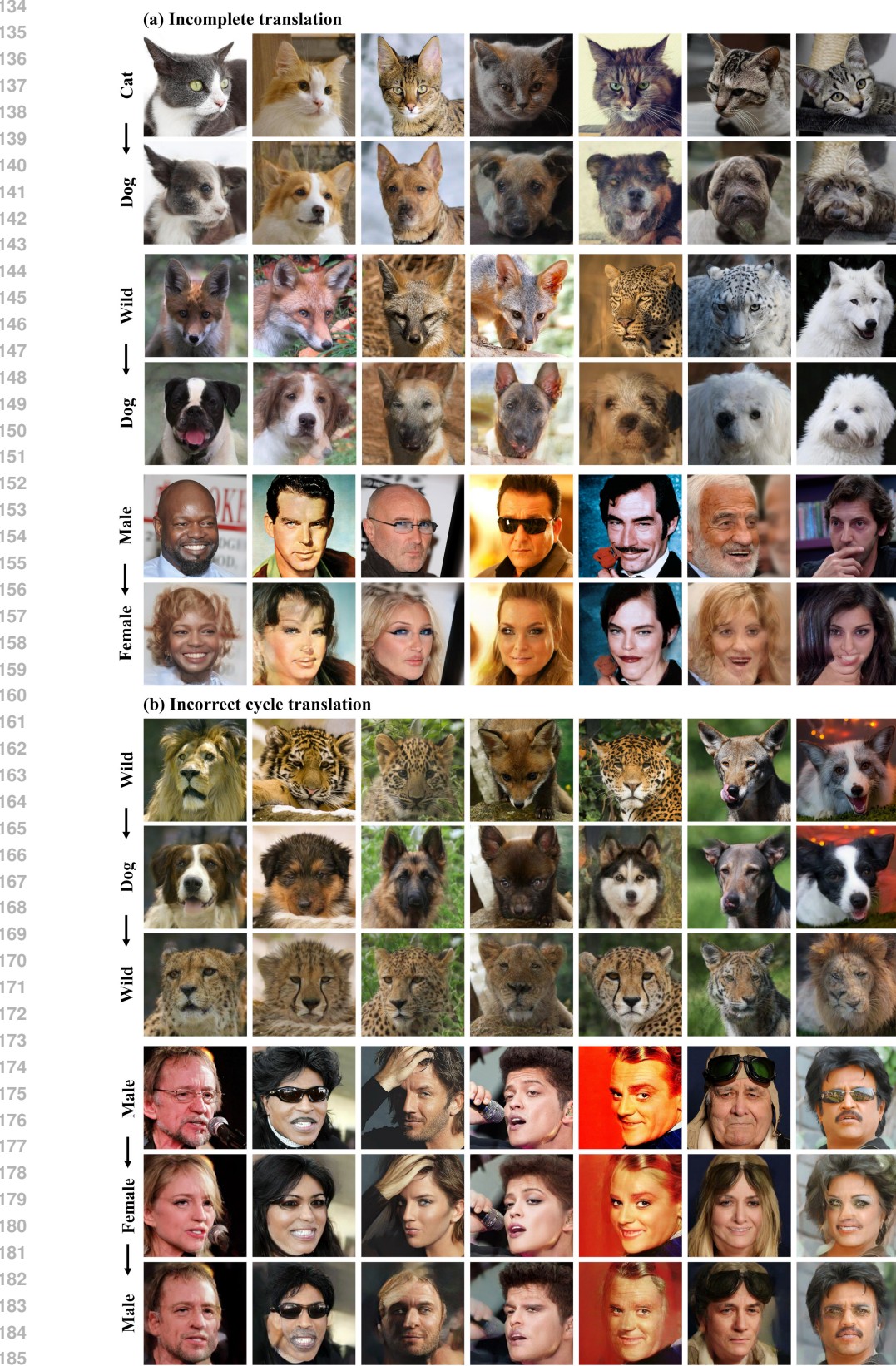

Figure 11: Example of failure cases, which are (a) incomplete translation and (b) incorrect cycle translation.

Table 6: **Quantitative comparison of unpaired image-to-image translation tasks (opposite & cycle translation)**. The opposition task used the same model and inference hyperparameters as the main direction task using bi-directionality.

| Method | NFE | FID↓ | PSNR ↑ | SSIM ↑ | Density ↑ | Coverage ↑ |
|---|---|---|---|---|---|---|
| **Dog→Cat** | | | | | | |
| StarGAN v2 (Choi et al., 2020) | 1 | 37.73 | 16.02 | 0.399 | 1.336 | 0.778 |
| CycleDiffusion (Wu & De la Torre, 2023) | 1000(+100) | 40.45 | 17.83 | 0.493 | 1.064 | 0.774 |
| DDIB (Teacher) (Su et al., 2023) | 160 | 30.28 | 17.15 | 0.597 | 2.071 | 0.902 |
| **IBCD (Ours)** | 1 | 28.99 | **19.10** | **0.695** | 1.699 | 0.894 |
| **IBCD† (Ours)** | 1 | **28.41** | 17.40 | 0.653 | **2.112** | **0.920** |
| **Dog→Wild** | | | | | | |
| StarGAN v2 (Choi et al., 2020) | 1 | 49.35 | 16.17 | 0.386 | 0.772 | 0.478 |
| CycleDiffusion (Wu & De la Torre, 2023) | 1000(+100) | 27.01 | 16.99 | 0.421 | 0.816 | 0.752 |
| DDIB (Teacher) (Su et al., 2023) | 160 | 13.20 | 16.80 | 0.583 | 1.202 | 0.760 |
| **IBCD (Ours)** | 1 | 18.79 | **17.56** | **0.671** | 0.900 | **0.830** |
| **IBCD† (Ours)** | 1 | **16.67** | 16.22 | 0.646 | **1.058** | 0.814 |
| **Female→Male** | | | | | | |
| StarGAN v2 (Choi et al., 2020) | 1 | 59.56 | 15.75 | 0.465 | 1.145 | 0.587 |
| DDIB (Teacher) (Su et al., 2023) | 160 | 26.98 | 18.74 | 0.668 | 1.154 | 0.858 |
| **IBCD (Ours)** | 1 | **31.28** | 19.93 | **0.733** | 1.300 | 0.808 |
| **IBCD† (Ours)** | 1 | 31.49 | **19.51** | 0.726 | **1.311** | **0.809** |
| **Cat→Dog→Cat** | | | | | | |
| StarGAN v2 (Choi et al., 2020) | 1 | 30.53 | 16.30 | 0.382 | 1.717 | 0.890 |
| CycleDiffusion (Wu & De la Torre, 2023) | 1000(+100) | 39.59 | 19.01 | 0.434 | 0.731 | 0.676 |
| DDIB (Teacher) (Su et al., 2023) | 160 | 16.56 | 25.88 | 0.804 | 1.330 | 0.990 |
| **IBCD (Ours)** | 1 | **22.42** | **22.35** | **0.767** | 1.322 | **0.992** |
| **IBCD† (Ours)** | 1 | 24.03 | 20.28 | 0.724 | **1.749** | 0.988 |
| **Wild→Dog→Wild** | | | | | | |
| StarGAN v2 (Choi et al., 2020) | 1 | 37.76 | 15.30 | 0.285 | 1.102 | 0.566 |
| CycleDiffusion (Wu & De la Torre, 2023) | 1000(+100) | 19.43 | 16.39 | 0.281 | 0.649 | 0.616 |
| DDIB (Teacher) (Su et al., 2023) | 160 | 6.75 | 26.08 | 0.803 | 1.118 | 0.974 |
| **IBCD (Ours)** | 1 | **9.89** | **20.56** | **0.739** | 1.118 | **0.972** |
| **IBCD† (Ours)** | 1 | 10.66 | 18.80 | 0.693 | **1.259** | 0.968 |
| **Male→Female→Male** | | | | | | |
| StarGAN v2 (Choi et al., 2020) | 1 | 57.80 | 15.39 | 0.502 | 1.634 | 0.728 |
| DDIB (Teacher) (Su et al., 2023) | 160 | 28.29 | 27.70 | 0.853 | 0.821 | 0.993 |
| **IBCD (Ours)** | 1 | **39.84** | **22.22** | **0.790** | **1.341** | 0.979 |
| **IBCD† (Ours)** | 1 | 39.96 | 21.85 | 0.783 | 1.332 | **0.984** |

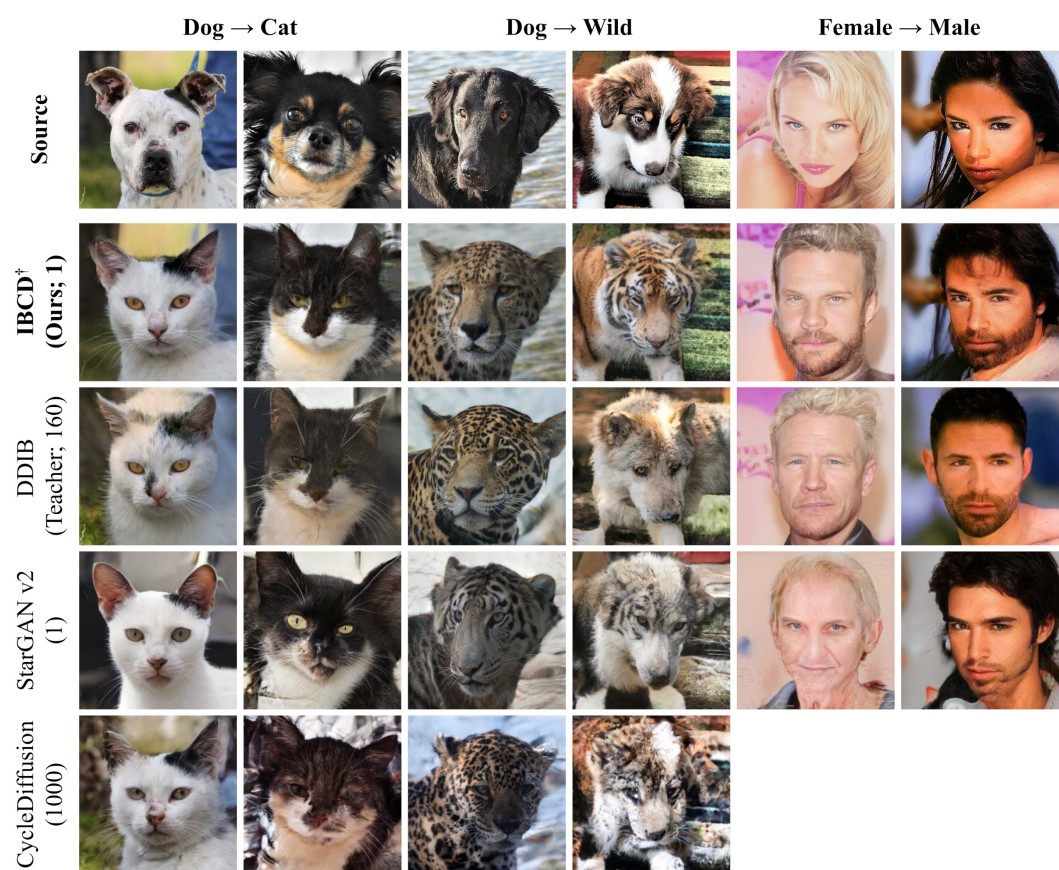

Figure 12: **Qualitative comparison of unpaired image-to-image translation tasks (opposite translation).** Compared to other baselines, our model achieves more realistic and source-faithful translations in a single step. The numbers in parentheses represent inference NFE.

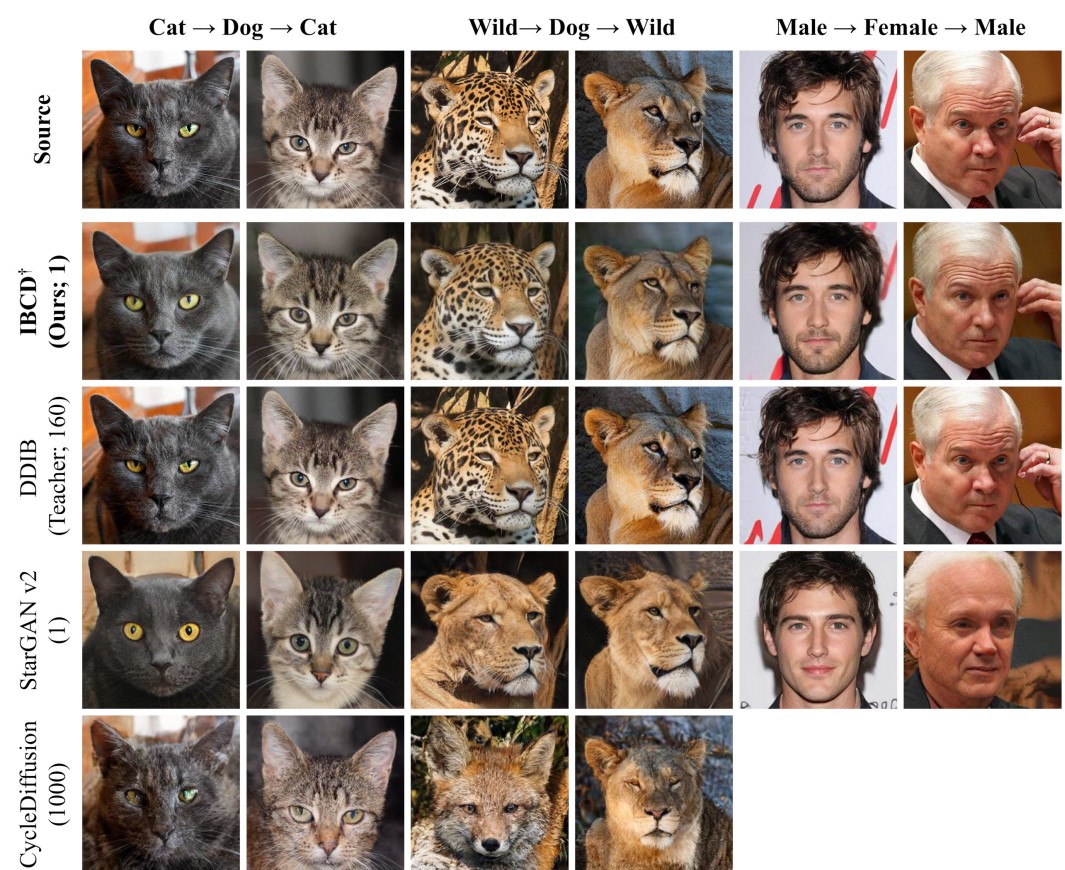

Figure 13: **Qualitative comparison of unpaired image-to-image translation tasks (cycle translation).** Compared to other baselines, our model achieves consistent cycle translations in a single step. The numbers in parentheses represent inference NFE.

**Cat** ⟶ **Dog** ⟶ **Cat**   **Dog** ⟶ **Cat** ⟶ **Dog**

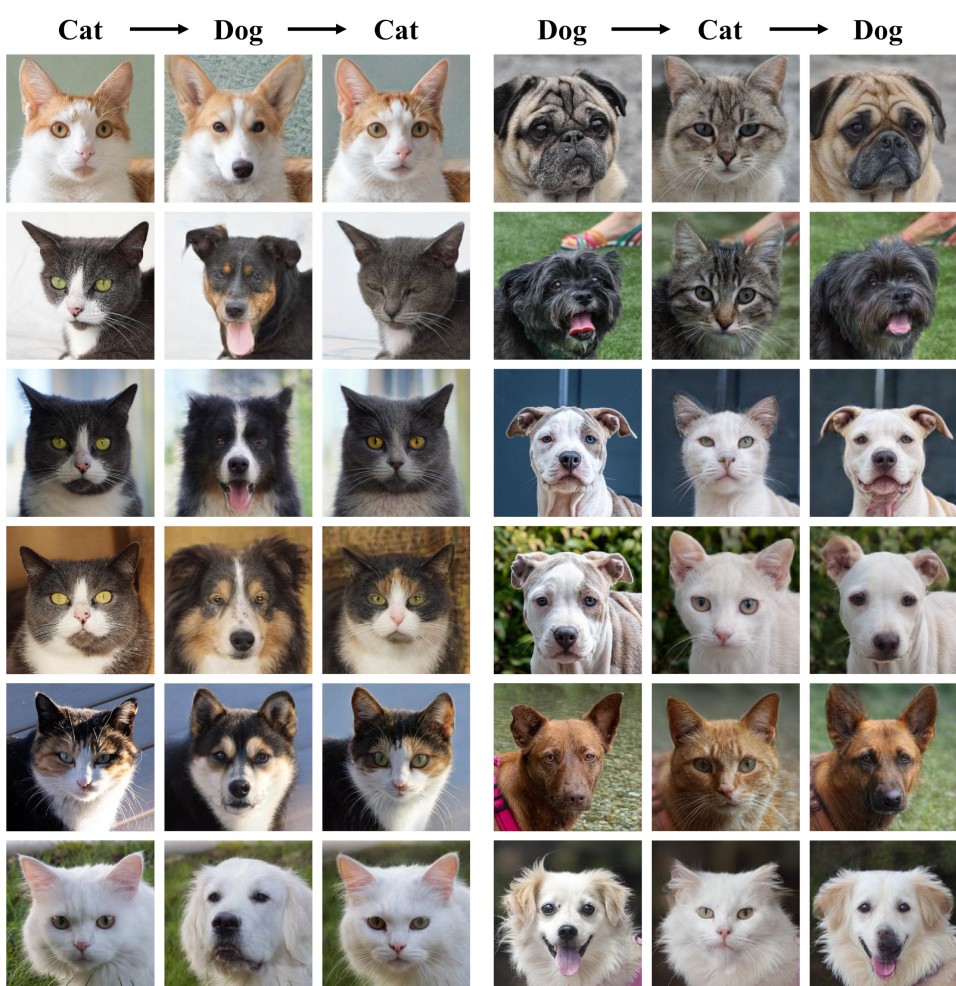

Figure 14: Result of the bi-directional cycle translation with a single model for the Cat↔Dog task (IBCD[†]).

**Wild** ⟶ **Dog** ⟶ **Wild**          **Dog** ⟶ **Wild** ⟶ **Dog**

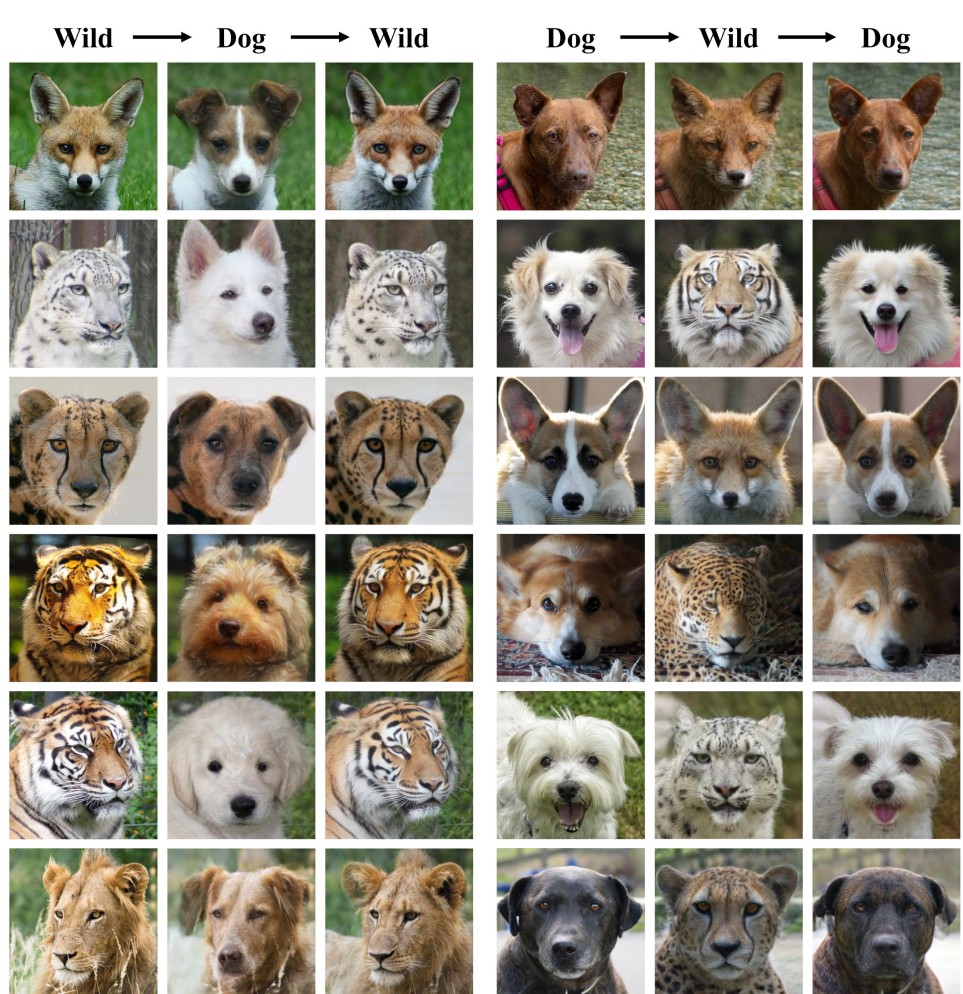

Figure 15: Result of the bi-directional cycle translation with a single model for the Wild↔Dog task (IBCD†).

**Male ⟶ Female ⟶ Male**  **Female ⟶ Male ⟶ Female**

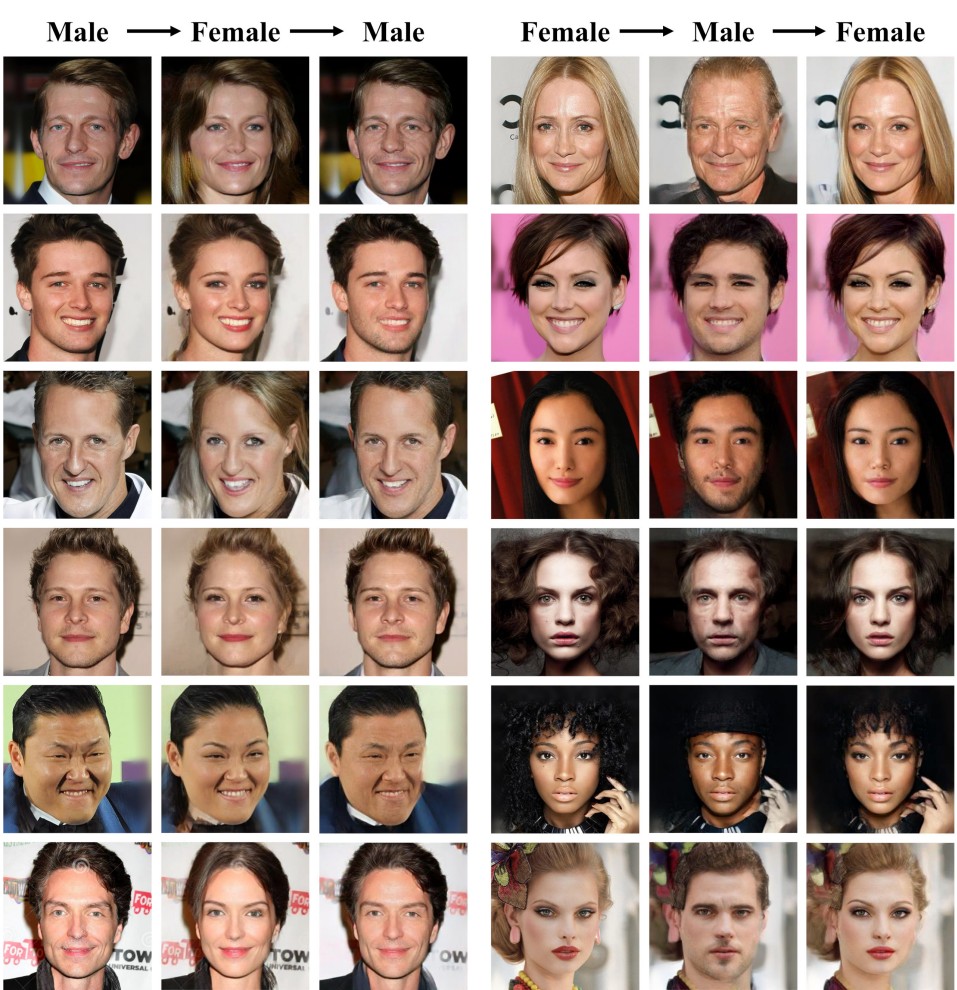

Figure 16: Result of the bi-directional cycle translation with a single model for the Male↔Female task (IBCD[†]).

