# OpenReview forum: "Implicit Bridge Consistency Distillation for One-Step Unpaired Image Translation"
_ICLR.cc/2025/Conference — Submitted to ICLR 2025_

### Official Review · Reviewer_etdT · 2024-10-26

**Soundness:** 4
**Presentation:** 3
**Contribution:** 3
**Rating:** 8
**Confidence:** 5

**Summary:**

The paper proposes to apply consistency distillation (CD) on previous DDIB, achieving a one-step generative model for unpaired image-to-image translation. The authors manage to extend the CD theory, which is applicable to two arbitrary distributions. The novel distribution matching and adaptive weighting techniques further stabilize and facilitate the training process. Both qualitative and quantitative experiments confirm the efficacy of the pipeline and the outperformance.

**Strengths:**

- The paper provides a versatile pipeline for unpaired image-to-image translation within only one step, which outperforms most previous methods even with large NFEs.

- The theory part is clear and intuitive, and the toy data showcases the instability of vanilla IBCD clearly.

- The experimental results are convincing and impressive, demonstrating directly the outperformance.

- The novel adaptive weighting is interesting and effective, encouraging further study in diffusion model distillation with insight.

**Weaknesses:**

- CD highly bases on PF-ODE, i.e., it needs to follow the score function trajectory. In Eq. (6), two PF-ODEs starting from different domains are connected together at $\sigma_{max}$, how to guarantee the smoothness of score function (i.e., gradient) at this point (since one directly uses noisy $x_a$ to solver attached to domain B)? If not smooth, how will the error be like? The authors may provide analysis here similar to original CD paper.

- In L261, the authors claim one of the challenge is to employ only local consistency. However, CTM [1] refers to local consistency as the case when applying PF-ODE solver with extremely smaller step. On the contrary, when using two adjacent timesteps, CTM names it global consistency, similar to original CD. So in the paper, this should also be called a global consistency. I can hardly understand why such strategy is a challenge, given that most distillation works use such a loss.

[1] Learning Probability Flow ODE Trajectory of Diffusion. Kim et al., ICLR 2024.

- The authors state that vanilla IBCD faces mean prediction phenomenon, but provides no convincing analysis on it. Original CD seems not to face such challenge. Does it come from the mismatch of two PF-ODEs? The visualization in Fig. 3(a) fails to convince me. The synthesized samples are not at the mean of domain B. Besides, I cannot see the efficacy of DMCD and cycle loss.

- The ablation study is somewhat confusing. Why vanilla IBCD is only a point rather than a broken line like the others? From Tab. 3 and Fig. 6, it seems that adaptive weighting may harm the performance, which is not consistent with conclusion in toy data. Conversely, DMCD is helpful in real data but fails in toy data. The authors may need further clarification.

**Questions:**

- The authors propose to use one generator for two domains, which may be unreasonable or hard to achieve in practice. I think the whole pipeline is compatible with two independent pre-trained DMs, i.e., one ADM on LSUN Cat and one ADM on ImageNet with some specific class.

---

> ### Author Response · Authors · 2024-11-22
>
> **W1. In Eq. (6), two PF-ODEs starting from different domains are connected together at $\sigma_{max}$, how to guarantee the smoothness of score function (i.e., gradient) at this point (since one directly uses noisy to solver attached to domain B)? If not smooth, how will the error be like?**
>
> We thank the reviewer for the insightful question. While IBCD introduces a continuous but non-differentiable point at the center of the PF-ODE trajectory ($t=0$), we argue that Theorem 1 of the original CD paper (Appendix A.2) remains applicable.
>
> First, we consider the Lipschitz condition for $f_\theta (x_t,t)$. Since the primary difference between IBCD and CD lies in the $t$ direction, we focus on this aspect. The output of $f_\theta$, which predicts the clean target domain image, remains constant along a given PF-ODE trajectory, regardless of $t$. Therefore, the Lipschitz condition is not affected by the non-differentiable point while the trajectory is continuous. Since the change in the $x_t$ direction is not different from CD, we can still use their Lipschitz assumption.
>
> Second, we examine the local truncation error of the ODE solver. The non-differentiable point is precisely captured by our discretization scheme. The gradient used at this point is a combination of gradients from both sides of the trajectory, ensuring stable numerical integration. For example, consider an Euler solver. For the forward direction (domain A to B):
> - The interval $i=[-1,0]$ uses the gradient at $-1$ (from domain A).
> - The interval $i=[0,1]$ uses the gradient at $0$ (from domain B).
>
> On the other hand, for the Backward direction (domain B to A):
> - The interval $i=[0,1]$ uses the gradient at $1$ (from domain B).
> - The interval $i=[-1,0]$ uses the gradient at $0$ (from domain A).
>
> Consequently, due to the properties of the consistency function and our careful handling of the non-differentiable point, the error bound of the consistency function remains $O((\Delta t)^p)$ which is same as the original CD.
>
> **W2. In L261, the authors claim one of the challenge is to employ only local consistency. However, this should be called a global consistency following [1]. I can hardly understand why such strategy is a challenge, given that most distillation works use such a loss.**
>
> In response to the reviewer's comment, we would like to clarify that the consistency loss employed in IBCD is indeed a form of **local consistency (vanilla CD)**, as defined in CTM. As extensively discussed in Section 5.2 and Appendix C.3 of CTM, this local consistency has key limitations, primarily its recursive nature.
>
> We argue that this recursive nature of local consistency increases distillation errors in IBCD, as we will explain later. However, we have also  revised our analysis of the sources of distillation errors in vanilla IBCD to adopt a more nuanced perspective:
>
> 1.	**Model Capacity Constraints**: Unlike other methods, our student model is tasked with learning a bidirectional translation function using a single model, inherently limiting its capacity. Previous work [2] has demonstrated that bidirectional consistency models can underperform compared to vanilla CM.
>
> 2.	**Combination of Different ODEs**: Unlike the Teacher model, where the two ODEs share a timestep and are differentiated solely by their initial conditions, the two ODEs in IBCD are entirely separate and interconnected. This can similarly impact model capacity and learning complexity.
>
> 3. **Recursive Nature of Local Consistency Loss**: As the local consistency loss is applied recursively, local fitting errors accumulate sequentially from the boundary condition to the trajectory's end. Consequently, the translation process, involving a longer path, incurs a larger accumulated distillation error compared to the generation process (updated Figure 8).
>
> **W3. The authors state that vanilla IBCD faces mean prediction phenomenon, but provides no convincing analysis on it. Original CD seems not to face such challenge.**
>
> Based on your review, we acknowledge the lack of theoretical support for the mean prediction phenomenon. While we initially introduced this phenomenon to illustrate an example of distillation error in vanilla IBCD, we have removed this content from the main text. Instead, we now focus on why distillation error is an issue in vanilla IBCD (W2 and section 3.2).
>
> Our auxiliary losses were designed to address this fundamental distillation error and provide flexibility in balancing reality-faithfulness, not to specifically target the mean prediction phenomenon. Therefore, our core argument remains unaffected by this change.
>
> **W4. The ablation study is somewhat confusing. Why vanilla IBCD is only a point rather than a broken line like the others?**
>
> Per your suggestion, we've modified the vanilla IBCD model to also incorporate a trade-off curve format. The data point prior to the update signifies the initial state of the model before additional training with the auxiliary loss function.

---

> ### Author Response · Authors · 2024-11-22
>
> **W5. I cannot see the efficacy of DMCD and cycle loss in toy data.**
>
> All components contribute to the effectiveness of the toy experiment. When DMCD is applied alone, it reduces the number of samples translated to low-density regions, as evidenced by the decreased density between the spirals. However, this comes at the cost of reduced modal coverage, leading to thinner spiral arms. Introducing cycle loss helps to mitigate this issue by maintaining the reduction in low-density translations while expanding the modal coverage, resulting in wider spiral arms and lower density between spirals. The recovery in modal coverage with cycle loss can be attributed to the difficulty of mapping samples back to the source domain when they are translated to the same point (reduced modal coverage) in the target domain. Finally, adaptive DMCD further enhances the ability of DMCD to reduce low-density translations. In addition to mitigating distillation error, auxiliary losses provide a mechanism for flexible control over the trade-off between reality and faithfulness by allowing for the adjustment of weights like EGSDE (Figure 9).
>
>
> **W6. From Tab. 3 and Fig. 6, it seems that adaptive weighting may harm the performance, which is not consistent with conclusion in toy data.**
>
> We agree the potential negative impact of adaptive DMCD on the low reality-high faithfulness region. That said,  the adaptive DMCD is usually a usful option since we can  use adaptive DMCD when prioritizing FID and omitting it when SSIM is the primary concern.
>
> While a definitive analysis is still needed, we hypothesize that the interaction between cycle loss and adaptive DMCD might be the root cause. Cycle loss, by transforming teacher's ODE trajectories trained with consistency loss, could potentially disrupt the semantic interpretation of consistency loss magnitudes, which are crucial for adaptive DMCD weightings. High cycle loss might lead to trajectories that deviate significantly from the teacher's, potentially undermining the assumption that consistency loss magnitude correlates with trajectory learning difficulty. This speculative explanation suggests a promising direction for future research: adaptively applying cycle loss to mitigate conflicts between auxiliary losses.
>
>
> **Q1. The authors propose to use one generator for two domains, which may be unreasonable or hard to achieve in practice. I think the whole pipeline is compatible with two independent pre-trained DMs.**
>
> As you pointed out, our pipeline can be compatible with two independent teacher models with a simple modification. In this case, it can be used as is even if the two domains are not trained in a single model, which expands the scope of our framework. Thanks for the great comment.
>
>
> **References:**
>
> [1] Kim, Dongjun, et al. "Consistency trajectory models: Learning probability flow ode trajectory of diffusion." *ICRL* (2024).
>
> [2] Li, Liangchen, and Jiajun He. "Bidirectional Consistency Models." arXiv preprint (2024).

---

> ### Author Response · Authors · 2024-11-25
>
> Dear reviewer etdT,
>
> We've carefully considered your valuable feedback and have made the following revisions to our manuscript and responses to your questions:
>
> 1. Clarified the error induced by non-differentiable ODE trajectories.
> 2. Clarified the difficulty in vanilla IBCD and removed the term "mean prediction phenomenon."
> 3. Clarified the ablation studies.
>
> As the discussion period deadline is approach fast, we would appreciate it if you could provide your feedback whether our revision and rebuttal have fully addressed your concerns.
>
> We appreciate your time and consideration.
>
> Best regards, Authors

---

> > ### Comment · Reviewer_etdT · 2024-11-25
> >
> > Thanks for the authors for the efforts and feedbacks, I think most of my questions are addressed by the rebuttal, so I raise my score accordingly. I would appreciate if the authors could add the results in Q1. in the revision.
> >
> > Besides, as for the diversity or degraded stochasticity mentioned by Reviewer UgkH, I think it could be a solution to add a random perturbation at $\sigma_{max}$, *i.e.*, manually increase stochasticity at the intersection between two domains.

---

> ### Author Response · Authors · 2024-11-25
>
> Thank you for your kind words and for taking the time to review our work. We appreciate your understanding and the increase in your score.
>
> As you suggested, we will include the results for Q1 in the revised manuscript. We also thank you for the insightful suggestion regarding diversity.
>
> Thank you again for your valuable feedback.

---

### Official Review · Reviewer_CDyb · 2024-11-01

**Soundness:** 3
**Presentation:** 2
**Contribution:** 2
**Rating:** 5
**Confidence:** 3

**Summary:**

This paper proposes a framework called Implicit Bridge Consistency Distillation (IBCD) for unpaired image to image translation. IBCD connects PF-ODE trajectories from any distribution to another one by extending consistency distillation with a diffusion implicit bridge model. It introduces Distribution Matching for Consistency Distillation (DMCD) and distillation-difficulty adaptive weighting method to deal with the distillation errors and mean prediction problems from the consistency distillation. Experiments on translation benchmarks demonstrate the effectiveness of the proposed method.

**Strengths:**

1) The paper is well-written and it clearly explains the proposed method.
2) The visualizations of component’s cumulative contributions on the toy dataset in Fig. (3) help appreciate the role of each part.
3) Experiments on both toy and highdimensional datasets demonstrate the effectiveness of IBCD.

**Weaknesses:**

1) Missing comparison of the results for bidirectional translation.
2) Missing comparison of computation cost with the existing methods to show the efficiency of the proposed method.
3) The results in Tab. 3 show the model which added DMCD loss, cycle loss and adaptive DMCD degrades the performance in terms of PSNR and SSIM compared to the method using IBCD only.
4) The zero in the first row of Eq. (6) might be \chi_A\cap\chi_B.

**Questions:**

see weakness

---

> ### Author Response · Authors · 2024-11-22
>
> **W1. Missing comparison of the results for bidirectional translation.**
>
> We appreciate the reviewer's suggestion to include quantitative comparative experiments on bidirectional translation. In response, we have added a comprehensive evaluation in Appendix D.6, where we conduct opposite translation (Dog→Cat, Dog→Wild, Female→Male) and cycle translation (Cat→Dog→Cat, Wild→Dog→Wild, Male→Female→Male) tasks. While bidirectional models and public checkpoints are limited, our results in **Table 6, Figures 12 and 13** demonstrate that our model's bidirectional performance is on par with its unidirectional capabilities.
>
> **(Partial) Table 6: Quantitative comparison of unpaired image-to-image translation tasks (opposite & cycle translation)**
> | Task  | Method | FID $\downarrow$ | PSNR $\uparrow$ | SSIM $\uparrow$ | Density  $\uparrow$ | Coverage  $\uparrow$ |
> |---------------|----------------|:---------------:|:---------------:|:------------------:|:----------------:|:----------------:|
> |Dog$\rightarrow$Cat|StarGAN v2 | 37.73 | 16.02 | 0.399 | 1.336 | 0.778 |
> ||CycleDiffusion | 40.45 | 17.83 | 0.493 | 1.064 | 0.774 |
> || **IBCD (Ours)** | 28.99 | **19.10** | **0.695** | 1.699 | 0.894 |
> || **IBCD$\dagger$ (Ours)** | **28.41** | 17.40 | 0.653 | **2.112** | **0.920** |
> |Cat$\rightarrow$Dog$\rightarrow$Cat|StarGAN v2 | 30.53 | 16.30 | 0.382 | 1.717 | 0.890 |
> || CycleDiffusion | 39.59 | 19.01 | 0.434 | 0.731 | 0.676 |
> || **IBCD (Ours)** | **22.42** | **22.35** | **0.767** | 1.322 | **0.992** |
> || **IBCD$\dagger$ (Ours)** | 24.03 | 20.28 | 0.724 | **1.749** | 0.988 |
> |...|...|...|...|...|...|...|
>
>
> **W2. Missing comparison of computation cost with the existing methods to show the efficiency of the proposed method.**
>
> Per your suggestion, we have conducted additional experiments comparing the actual inference time of our method with major open-source baselines. As shown in **(new) Table 5 and Appendix D.4**, our methodology demonstrates a significant advantage in inference computational complexity.
>
> **(Partial) Table 5: Quantitative comparison of model inference times.**
> | Method         |...| Time [s/img] $\downarrow$ | Relative Time $\downarrow$ |
> |----------------|:-:|:------------:|:-------------:|
> | StarGan v2     |...|     0.058    |      5.5      |
> | CUT            |...|     0.068    |      6.4      |
> | UNSB           |...|     0.104    |      9.9      |
> | ILVR           |...|    12.915    |     1224.2    |
> | SDEdit         |...|     6.378    |     604.5     |
> | EGSDE          |...|    15.385    |     1458.3    |
> | CycleDiffusion |...|    26.032    |     2467.5    |
> | DDIB (Teacher) |...|     0.965    |      90.6     |
> | **IBCD (Ours)**|...|   **0.011**  |     **1**     |
>
>
> **W3. The results in Table 3 show the model which added DMCD loss, cycle loss and adaptive DMCD degrades the performance in terms of PSNR and SSIM compared to the method using IBCD only.**
>
> To clarify the comparison, we've updated Table 3 to benchmark against the lowest FID achievable by each individual component. Additionally, we've introduced a new metric, PSNR-Teacher, which uses the DDIB teacher's output as a PSNR ground truth. This metric allows us to assess the resolution of distillation error, a primary focus of our auxiliary loss approach.
>
> Our results demonstrate that each added component consistently reduces FID beyond the lower bound achievable by vanilla IBCD, while effectively mitigating the inherent PSNR trade-off and minimizing distillation error. Notably, updated Figure 6 highlights that even under identical PSNR conditions, the application of auxiliary losses consistently yields lower (improved) FID scores or vice versa.
>
> **Table 3: Quantitative ablation study results in the Cat→Dog task under the lowest FID.**
> | Component            | FID $\downarrow$ | PSNR-teacher $\uparrow$ | PSNR-source $\uparrow$ |
> |----------------------|:----------------:|:-----------------------:|:----------------------:|
> | IBCD only            | 48.12            | 18.27                   | 19.02                  |
> | + DMCD               | 44.40            | 17.95                   | 16.80                  |
> | + DMCD & Cycle       | 44.31            | 18.22                   | 17.19                  |
> | + adap. DMCD & Cycle | 44.69            | 18.97                   | 18.04                  |
>
>
> **W4. The zero in the first row of Eq. (6) might be $\chi_A \cap \chi_B$.**
>
> Thanks for your careful reading. Typo fixed.

---

> > ### Comment · Reviewer_CDyb · 2024-11-26
> >
> > I thank the authors for addressing my concerns. But I lean to agree with reviewer UgkH about the approach's combination nature.

---

> > > ### Author Response · Authors · 2024-11-27
> > >
> > > We appreciate the reviewer's acknowledgment of our efforts to address the initial concerns. While our framework leverages multiple techniques, it does so in a novel and non-trivial manner, pushing the boundaries of current methodologies. Our key contributions are as follows:
> > >
> > > 1. **Novel Framework**: Our model uniquely satisfies four critical properties—*one-step, unpaired, bidirectional, and non-discriminator*—simultaneously. To our knowledge, no prior work has achieved this combination.
> > >
> > > 2. **Technical Innovations**: Extending consistency distillation to *bidirectional diffusion bridge* models required significant technical advancements. Our contributions include:
> > >
> > > - Adapting consistency distillation to bidirectional & bridge trajectories by innovating on timestep design, training schemes, boundary conditions, and model parametrization.
> > >
> > > - Introducing auxiliary losses (e.g., DMCD, adaptive DMCD, bidirectional cycle loss) that were uniquely integrated and modified for our IBCD framework to reduce distillation loss, and enhance reality and fidelity.
> > >
> > > 3. **Validated Superiority**: Extensive experiments, ranging from toy to many high-dimensional datasets, validate the significant impact of these innovations on performance across diverse tasks and metrics, establishing new state-of-the-art results. Another reviewer has also acknowledged the significance of this contribution.
> > >
> > > We believe our work offers a valuable contribution to the field and hope this additional explanation clarifies its novelty and significance.
> > >
> > > We are open to further feedback and suggestions to improve the manuscript. Thank you for taking the time to review our work.

---

> ### Author Response · Authors · 2024-11-25
>
> Dear reviewer CDyb,
>
> We've carefully considered your valuable feedback and have made the following revisions to our manuscript and responses to your questions:
>
> 1. Added bidirectional translation evaluation results.
> 2. Added inference computational cost comparison results.
> 3. Clarified ablation studies.
>
> As the discussion period deadline is approach fast, we would appreciate it if you could provide your feedback whether our revision and rebuttal have fully addressed your concerns.
>
> We appreciate your time and consideration.
>
> Best regards, Authors

---

### Official Review · Reviewer_UgkH · 2024-11-04

**Soundness:** 2
**Presentation:** 2
**Contribution:** 1
**Rating:** 3
**Confidence:** 1

**Summary:**

Diffusion models are widely used for image translation. This paper identifies limitations in existing approaches: slow inference, need for paired data, and one-way translation constraints. It introduces Implicit Bridge Consistency Distillation (IBCD) to address these issues. IBCD extends consistency distillation with a diffusion implicit bridge model. The paper proposes two improvements: Distribution Matching for Consistency Distillation (DMCD) and a distillation-difficulty adaptive weighting method. Experimental results show IBCD achieves state-of-the-art performance in one-step generation for bidirectional translation.

**Strengths:**

Pro:

- The sampling speed in image-to-image translation is a critical problem in this area.
- The paper combines various techniques, including DDIB and consistency models.

**Weaknesses:**

Con:

- The main concern is that the method seems too incremental, appearing to be merely a combination of DDIB and consistency models.
- In Table 3, the FID improvement from adding Cycle and DMCD is marginal. Is the author aware of what a 0.1 FID change means? If you repeat the experiment twice, the variance might be even larger. This becomes a significant issue when the FID is so high. Also, most baselines in Table 2 show the variance of FID, while the author didn't. As you can see, the variance of other methods is quite large, further undermining the ablation study in Table 3.
- With only a single step, the stochasticity is significantly reduced. The authors should include several other related metrics that highlight diversity, such as the Inception Score. Additionally, more failure cases should be provided for better understanding of the method's limitations.

**Questions:**

as above

---

> ### Author Response · Authors · 2024-11-22
>
> **W1: The method seems too incremental, appearing to be merely a combination of DDIB and consistency models.**
>
> In contrast to your misunderstanding, our work introduces a novel image translation model framework that uniquely satisfies four crucial properties: one-step, unpaired, bidirectional, and non-discriminator. This framework establishes a new state-of-the-art performance across various tasks and evaluation metrics while maintaining these desirable properties.
>
> Extending consistency distillation to DDIB, particularly in a bidirectional manner, has never been tried before due to several technique challenges.  To address this, we have developed several novel approaches such as adapting consistency distillation to diffusion bridge trajectories (timesteps, training scheme), designing appropriate boundary conditions for bidirectional training, and carefully parameterizing the model for bidirectional diffusion bridge (Appendix B). Furthermore, we introduced additional auxiliary losses, strategically designed to mitigate the distillation loss and simultaneously improve reality and fidelity. These losses were carefully integrated into the IBCD framework with framework-aware modification (DMCD, adaptive DMCD, bidirectional cycle loss).
>
> Extensive ablation studies validate the significance of these advances in achieving our framework's superior performance across various tasks and evaluation metrics. Therefore, our work cannot be considered as incremental.
>
>
> **W2. In Table 3, the FID improvement from adding Cycle and DMCD is marginal.**
>
> As you correctly noted, the bottom three components of Table 3 do **not directly compare the superiority of FID**. Instead, they present a scenario where PSNR is considered in situations with similar FID values.
>
> To clarify the comparison, we've updated Table 3 to benchmark against the lowest FID achievable by each individual component. Additionally, we've introduced a new metric, PSNR-Teacher, which uses the DDIB teacher's output as a PSNR ground truth. This metric allows us to assess the resolution of distillation error, a primary focus of our auxiliary loss approach.
>
> Table 3 demonstrates that each added component consistently reduces FID beyond the lower bound achievable by vanilla IBCD, while effectively mitigating the inherent PSNR trade-off and minimizing distillation error. Notably, updated Figure 6 highlights that even under identical PSNR conditions, the application of auxiliary losses consistently yields lower FID scores or vice versa.
>
> **Table 3: Quantitative ablation study results in the Cat→Dog task under the lowest FID.**
> | Component            | FID $\downarrow$ | PSNR-teacher $\uparrow$ | PSNR-source $\uparrow$ |
> |-|:-:|:-:|:-:|
> | IBCD only            | 48.12            | 18.27        | 19.02    |
> | + DMCD               | 44.40            | 17.95                   | 16.80                  |
> | + DMCD & Cycle       | 44.31            | 18.22                   | 17.19                  |
> | + adap. DMCD & Cycle | 44.69            | 18.97                   | 18.04                  |
>
> **W3. Most baselines in Table 2 show the variance of FID, while the author didn't.**
>
> The criteria for indicating and not indicating the variance of the results are the same as those adopted in many previous studies. Models that indicate variance in quantitative results (ILVR, SDEdit, EGSDE, SDDM) utilize a probabilistic sampler (SDE), generating probabilistic samples for each translation. Consequently, their variance is explicitly indicated. In contrast, models that employ non-deterministic sampling during the translation process (ours, CycleDiffusion, GAN-based models, etc.), do not indicate variance as there is **no inherent probabilistic mechanism in the translation process**.
>
> **W4. With only a single step, the stochasticity is significantly reduced. The authors should include several other related metrics that highlight diversity, such as the Inception Score.**
>
> To address the reviewer's concern, we additionally conducted a quantitative diversity analysis using the **density-coverage metric**[1] in addition to the FID metric (updated Table 2). This metric, which separately evaluates quality and diversity, is more suitable for our tasks, as Inception Score's (Inception v3) classification granularity is insufficient to assess diversity within a single domain (dog or female).
>
> As shown in the **Table 2 (please refer to the updated text)**, our one-step non-deterministic mapping approach outperforms baselines in both quality (density) and diversity (coverage).
>
> **W5. More failure cases should be provided for better understanding of the method's limitations.**
>
> Per your suggestion, we have updated Figure 11 to include failure cases for each task, providing a more comprehensive visualization of our model's performance (14 → 35 cases).
>
> **References:**
>
> [1] Naeem, Muhammad Ferjad, et al. "Reliable fidelity and diversity metrics for generative models." *ICML* (2020).

---

> ### Author Response · Authors · 2024-11-25
>
> Dear reviewer UgkH,
>
> We've carefully considered your valuable feedback and have made the following revisions to our manuscript and responses to your questions:
>
> 1. Clarified our contributions and the determinism of the inference process.
> 2. Clarified the ablation studies.
> 3. Added diversity evaluation (density-coverage) results.
> 4. Added extra failure cases.
>
> As the discussion period deadline is approach fast, we would appreciate it if you could provide your feedback whether our revision and rebuttal have fully addressed your concerns.
>
> We appreciate your time and consideration.
>
> Best regards, Authors

---

> ### Author Response · Authors · 2024-11-26
>
> This is just a kind reminder as the deadline for the paper revision period is approaching.  We are looking forward to hearing your feedback and will be happy to answer any further questions.

---

> ### Author Response · Authors · 2024-11-28
>
> Dear Reviewer UgkH,
>
> As the discussion deadline is getting closer, we would like to kindly remind the reviewer that we are waiting for your valuable feedback to our responses.
>
> Please note that we've addressed all your feedback by **clarifying our contributions**, **ablation studies**, and adding **diversity evaluation** and **failure cases**.
>
> So we would greatly appreciate it if you could review our revised manuscript and provide your feedback.
>
> Thank you for your time and consideration.
>
> Best,
> Authors

---

### Author Response · Authors · 2024-11-22
**General Response**

We thank the reviewers for their insightful comments and suggestions. We are pleased that the reviewers found our paper well-written, theoretically grounded, and clear, particularly in the validation of the toy experiment. We have carefully addressed all the points raised and made corresponding revisions to the manuscript, with corrections and additions highlighted in blue.

**Key changes:**
- Added density-coverage metric to main experiement for a more comprehensive evaluation (Table 2)
- Update ablation studies (Table 3, Figure 6).
- Clarified the necessity of the auxiliary loss in vanilla IBCD (Section 3.2).
- Added quantitative comparison of real-world inference speeds (Appendix D.4, Table 5).
- Included quantitative comparison of bidirectional tasks (Appendix D.6, Table 6, Figures 12, 13).
- Added more failure cases (Figure 11).

We will respond to each of the reviewer's comments individually, providing detailed responses to the reviewer's concerns. We hope that the revised manuscript and responses address the reviewer's concerns and provide a valuable contribution to the field. If you have any further questions, please feel free to discuss them at any time.

---

### Meta-Review · Area_Chair_117y · 2024-12-21

**Metareview:**

This paper proposes a method for unpaired image-to-image translation. The key idea is to leverage consistency distillation on an implicit bridge model. Overall, the reviewers appreciate the task and the visualization of the paper. However, the reviewers have concerns over the contribution/novelty of the paper, and the significance of the empirical results. In particular, two reviewers expressed that the approach is indeed a combination and remained unconvinced after the discussion period.

**Additional Comments On Reviewer Discussion:**

During the discussion period, most of the questions regarding details and clarification have been addressed. While reviewer CDyb mentioned that part of the concerns were addressed, the reviewer also mentioned the combined nature of the approach and did not raise the rating. Next, unfortunately, Reviewer UgkH did not engage during the discussion period. In this case, the AC has checked over the authors' response and found that it would be unlikely that Reviewer UgkH would change the opinion. For example,  Reviewer UgkH asked about the significance of the FID, stating ``If you repeat the experiment twice, the variance might be even larger.'' On the other hand, the authors responded with "do not indicate variance as there is no inherent probabilistic mechanism". This response misses the point, Reviewer UgkH is asking about the randomness from "repeat the experiment twice", e.g., if one were to train the model twice (with different random seeds) the result would not be the same. Next, the concerns about contribution and whether work is incremental is always a subjective matter. From the AC's perspective, the combination nature of an approach does not directly equate to a lack of contribution or novelty. However, the AC finds the motivation of such a combination to be a bit weak in the paper. Finally, the paper could be strengthened by conducting a user study on the improvement and providing more qualitative comparisons in the paper.

---

### Decision · Program_Chairs · 2025-01-22

Reject